# Indicators of Attack Failure: Debugging and Improving Optimization of Adversarial Examples

**Maura Pintor**
University of Cagliari, Italy
Pluribus One, Italy
maura.pintor@unica.it

**Luca Demetrio**
University of Genoa, Italy
Pluribus One, Italy
luca.demetrio@unige.it

**Angelo Sotgiu**
University of Cagliari, Italy
angelo.sotgiu@unica.it

**Ambra Demontis**
University of Cagliari, Italy
ambra.demontis@unica.it

**Nicholas Carlini**
Google
nicholas@carlini.com

**Battista Biggio**
University of Cagliari, CINI, Italy
Pluribus One, Italy
battista.biggio@unica.it

**Fabio Roli**
University of Genoa, CINI, Italy
Pluribus One, Italy
fabio.roli@unige.it

## Abstract

Evaluating robustness of machine-learning models to adversarial examples is a challenging problem. Many defenses have been shown to provide a false sense of robustness by causing gradient-based attacks to fail, and they have been broken under more rigorous evaluations. Although guidelines and best practices have been suggested to improve current adversarial robustness evaluations, the lack of automatic testing and debugging tools makes it difficult to apply these recommendations in a systematic manner. In this work, we overcome these limitations by: (i) categorizing attack failures based on how they affect the optimization of gradient-based attacks, while also unveiling two novel failures affecting many popular attack implementations and past evaluations; (ii) proposing six novel *indicators of failure*, to automatically detect the presence of such failures in the attack optimization process; and (iii) suggesting a systematic protocol to apply the corresponding fixes. Our extensive experimental analysis, involving more than 15 models in 3 distinct application domains, shows that our indicators of failure can be used to debug and improve current adversarial robustness evaluations, thereby providing a first concrete step towards automatizing and systematizing them. Our open-source code is available at: https://github.com/pralab/IndicatorsOfAttackFailure.

## 1 Introduction

Despite their unprecedented success in many different applications, machine-learning models have been shown to be vulnerable to *adversarial examples* [40, 4], i.e., inputs intentionally crafted to mislead such models at test time. While hundreds of defenses have been proposed to overcome this issue [46, 29, 43, 36], many of them turned out to be ineffective, as their robustness to adversarial inputs was significantly overestimated. In particular, some defenses were evaluated by running existing attacks with inappropriate hyperparameters (e.g., using an insufficient number of iterations), while others were implicitly *obfuscating* gradients, thereby causing the optimization of gradient-based attacks to fail. Such defenses have been shown to yield much lower robustness under more rigorous robustness evaluations and using attacks carefully adapted to evade them [10, 2].

36th Conference on Neural Information Processing Systems (NeurIPS 2022).

To prevent such evaluation mistakes, and help develop better defenses, evaluation guidelines and best practices have been described in recent work [11]. However, they have been mostly neglected as their application is non-trivial; in fact, 13 defenses published after the release of these guidelines were found again to be wrongly evaluated, reporting overestimated adversarial robustness values [41]. It has also been shown that, even when following such guidelines, robustness can still be overestimated [33]. Finally, while recently-proposed frameworks combining diverse attacks seem to provide more reliable adversarial robustness evaluations [13, 44], it is still unclear whether and to what extent the considered attacks can also be affected by subtle failures.

To address these limitations, in this work we propose the first *testing* approach aimed to debug and *automatically* detect misleading adversarial robustness evaluations. The underlying idea, similar to traditional software testing, is to lower the entry barrier for researchers and practitioners towards performing more reliable robustness evaluations. To this end, we provide the following contributions: (i) we categorize four known attack failures by connecting them to the optimization process of gradient-based attacks, which enabled us also to identify two additional, never-before-seen failures affecting many popular attack implementations and past evaluations (Sect. 2.1); (ii) we propose six *indicators of attack failures* (IoAF), i.e., quantitative metrics derived by analyzing the optimization process of gradient-based attacks, which can automatically detect the corresponding failures (Sect. 2.2); and (iii) we suggest a systematic, semi-automated protocol to apply the corresponding fixes (Sect. 2.3). We empirically validate our approach on three distinct application domains (images, audio, and malware), showing how recently-proposed, wrongly-evaluated defenses could have been evaluated correctly by monitoring the IoAF values and following our evaluation protocol (Sect. 3). We also open source our code and data to enable reproducibility of our findings at `https://github.com/pralab/IndicatorsOfAttackFailure`. We conclude by discussing related work (Sect. 4), limitations, and future research directions (Sect. 5).

## 2 Debugging Adversarial Robustness Evaluations

We introduce here our *automated testing* approach for adversarial robustness evaluations, based on the design of novel metrics, referred to as *Indicators of Attack Failure (IoAF)*, each aimed to detect a specific failure within the optimization process of gradient-based attacks.

### 2.1 Optimizing Gradient-based Attacks

Before introducing the IoAF, we discuss how gradient-based attacks are optimized, and provide a generalized attack algorithm that will help us to identify the main attack failures. In particular, we consider attacks that solve the following optimization problem:

$$\min_{\boldsymbol{\delta}} \quad L(\boldsymbol{x} + \boldsymbol{\delta}, y_t; \boldsymbol{\theta}), \tag{1}$$

$$\text{s.t.} \quad \|\boldsymbol{\delta}\|_p \leq \epsilon, \text{ and } \boldsymbol{x} + \boldsymbol{\delta} \in [0, 1]^d, \tag{2}$$

where $\boldsymbol{x} \in [0, 1]^d$ is the $d$-dimensional input sample, $y_t \in \mathcal{Y} = \{1, \ldots, c\}$ is the target class label (chosen to be different from the true label $y$ of the input sample), $\boldsymbol{\delta} \in \mathbb{R}^d$ is the perturbation applied to the input sample during the optimization, and $\boldsymbol{\theta}$ are the parameters of the model. The loss function $L$ is defined such that minimizing it amounts to having the perturbed sample $\boldsymbol{x} + \boldsymbol{\delta}$ misclassified as $y_t$. Typical examples include the Cross-Entropy (CE) loss, or the so-called *logit* loss [10], i.e., $L(\boldsymbol{x} + \boldsymbol{\delta}, y_t, \boldsymbol{\theta}) = \max_{k \neq y_t} f_k(\boldsymbol{x} + \boldsymbol{\delta}, \boldsymbol{\theta}) - f_{y_t}(\boldsymbol{x} + \boldsymbol{\delta}, \boldsymbol{\theta})$, being $f_i(\cdot, \boldsymbol{\theta})$ the model's prediction (*logit*) for class $i$.[1] Let us finally discuss the constraints in Eq. (2). While the $\ell_p$-norm constraint $\|\boldsymbol{\delta}\|_p \leq \epsilon$ bounds the maximum perturbation size, the box constraint $\boldsymbol{x} + \boldsymbol{\delta} \in [0, 1]^d$ ensures that the perturbed sample stays withing the given (normalization) bounds.

*Transfer Attacks.* When the target model is either non-differentiable or not sufficiently smooth [2], a surrogate model $\hat{\boldsymbol{\theta}}$ can be used to provide a smoother approximation of the loss $L$, and facilitate the attack optimization. The attack is thus optimized on the surrogate model, and then evaluated against the target model. If the attack is successful, then it is said to correctly *transfer* to the target [30].

---

[1]Note that, while Eq. (1) describes targeted attacks, untargeted attacks can be easily accounted for by substituting $y_t$ with $y$ and changing the sign of $L$.

**Algorithm 1:** Generalized gradient-based attack for optimizing adversarial examples.

**Input** : $\boldsymbol{x}$, the initial sample; $y_t$, the target class label; $n$, the number of iterations; $\alpha$, the step size; $\boldsymbol{\theta}$, the target model; $L$ the loss function; $\Pi$, the projection operator enforcing the constraints in Eq. (2).

**Output**: $\boldsymbol{x}^{\star}$, the solution found by the algorithm

1   $\boldsymbol{x}_0 \leftarrow \texttt{initialize}(\boldsymbol{x})$          ▷ Initialize starting point
2   $\hat{\boldsymbol{\theta}} \leftarrow \texttt{approximation}(\boldsymbol{\theta})$        ▷ Use surrogate model (if required)
3   $\boldsymbol{\delta}_0 \leftarrow \boldsymbol{0}$                ▷ Initialize $\delta$
4   **for** $i \in [1, n]$ **do**
5      $\boldsymbol{\delta}_i \leftarrow \boldsymbol{\delta}_{i-1} - \alpha \nabla_{\boldsymbol{x}} L(\boldsymbol{x} + \boldsymbol{\delta}_{i-1}, y_t; \hat{\boldsymbol{\theta}})$     ▷ Compute gradient update(s)
6      $\boldsymbol{\delta}_i \leftarrow \Pi(\boldsymbol{x}, \boldsymbol{\delta}_i)$      ▷ Project $\boldsymbol{\delta}$ onto the feasible domain (Eq. 2)
7   **return** $\boldsymbol{x}^{\star} \leftarrow \boldsymbol{x} + \texttt{best}(\boldsymbol{\delta}_0, ..., \boldsymbol{\delta}_n)$      ▷ Return best solution

**Generalized Attack Algorithm.** We provide here a generalized algorithm, given as Algorithm 1, which summarizes the main steps followed by gradient-based attacks to solve Problem (1)-(2). The algorithm starts by defining an *initialization point* (line 1), which can be the input sample $\boldsymbol{x}$, a randomly-perturbed version of it, or even a sample from the target class [5]. If the target model $\boldsymbol{\theta}$ is either non-differentiable or not sufficiently smooth, a surrogate model $\hat{\boldsymbol{\theta}}$ can be used to approximate it, and perform a transfer attack (line 2). The attack then iteratively updates the point to find an adversarial example (line 4), computing one (or more) gradient updates in each iteration (line 5), while the perturbation $\boldsymbol{\delta}_{i+1}$ is projected onto the feasible domain (i.e., the intersection of the constraints in Eq. 2) via a projection operator $\Pi$ (line 6). The algorithm finally returns the best perturbation across the whole *attack path*, i.e., the perturbed sample that evades the (target) model with the lowest loss (line 7). This generalized attack algorithm provides the basis for identifying known and novel attack failures, by connecting each failure to a specific step of the algorithm, as discussed in the next section.

**Other Perturbation Models.** We conclude this section by remarking that, even if we consider additive $\ell_p$-norm perturbations in our formulation, the proposed approach can be easily extended to more general perturbation models (e.g., represented as $\boldsymbol{x}' = h(\boldsymbol{x}, \boldsymbol{\delta})$, being $\boldsymbol{x}'$ the perturbed feature representation of a valid input sample, and $h$ a manipulation function parameterized by $\boldsymbol{\delta}$, as in [17]). In fact, our approach can be used to debug and evaluate any attack as long as it is optimized via gradient descent, regardless of the given perturbation model and constraints, as also demonstrated in our experiments (A.3,A.4, A.5).

## 2.2   Indicators of Attack Failure

We introduce here our approach, compactly represented in Fig. 1, by describing the identified attack failures, along with the corresponding indicators and mitigation strategies. We describe two main categories of failures, respectively referred to as *loss-landscape* and *attack-optimization* failures.

*Loss-landscape Failures, Mitigations, and Indicators.* The first category of failures depends on the choice of the loss function $L$ and of the target model $\boldsymbol{\theta}$, regardless of the specific attack implementation. In fact, it has been shown that the objective function in Eq. (1) may exhibit *obfuscated gradients* [10, 2], which prevent gradient-based attacks to find adversarial examples even when they exist within the feasible domain. We report two *known* failures linked to this issue, referred as $F_1$ and $F_2$ below.

**$F_1$: Shattered Gradients.** This failure is reported in [3, 8]. It compromises the computation of the input gradient $\nabla_{\boldsymbol{x}} L$, and the whole execution of the attack, when at least one of the model's components (e.g., a specific layer) is non-differentiable or causes numerical instabilities ($F_1$ in Fig. 3).

**$M_1$: Use BPDA.** This failure can be overcome using the Backward Pass Differentiable Approximation (BPDA) [3, 41], i.e., replacing the derivative of the problematic components with the identity matrix.

**$I_1$: Unavailable Gradients.** Despite the failure and mitigation being known, no automated, systematic approach has ever been proposed to detect $F_1$. This newly-proposed indicator is able to automatically detect the presence of non-differentiable components and numerical instabilities when computing gradients; in particular, for each given input $\boldsymbol{x}$, if the input gradient $\nabla_{\boldsymbol{x}} L(\boldsymbol{x}, y_t, \boldsymbol{\theta})$ has zero norm, or if its computation returns an error, we set $I_1(\boldsymbol{x}) = 1$ (and 0 otherwise).

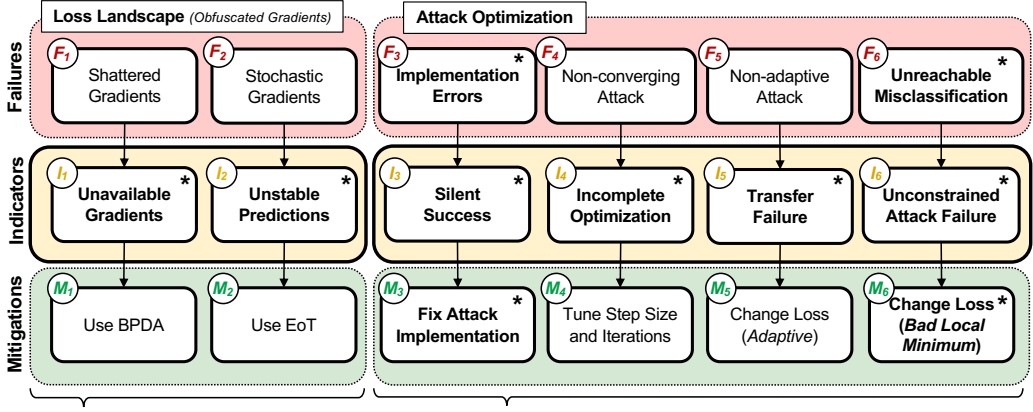

Figure 1: Indicators of Attack Failure (IoAF). We show the connections among the distinct attack failures (*top*), the proposed IoAF (*center*), and the corresponding mitigations (*bottom*). The novel failures, indicators, and mitigations found in this work are highlighted in bold and marked with "∗".

**$F_2$: Stochastic Gradients.** This failure is reported in [2, 41]. The input gradients can be computed, but their value is uninformative, as following the gradient direction does not correctly minimize the loss function $L$. Such issue arises when models introduce randomness in the inference process [22], or when their training induces noisy gradients [43] ($F_2$ in Fig. 3).

**$M_2$: Use EoT.** Previous work mitigates this failure using *Expectation over Transformation* (EoT) [2, 41], i.e., by averaging the loss $L$ over the same transformations performed by the randomized model, or over random perturbations of the input (sampled from a given distribution).

**$I_2$: Unstable Predictions.** Despite the failure and mitigation being known, no automated, systematic approach has been considered to automatically detect $F_2$. This novel indicator measures the relative variability $V(\boldsymbol{x})$ of the loss function $L$ around the input samples. Given an input sample $\boldsymbol{x}$, we draw $s$ perturbed inputs uniformly from a small $\ell_2$ ball centered on $\boldsymbol{x}$, with radius $r$, and compute $V(\boldsymbol{x}) = \min(\frac{1}{s}\sum_{i=1}^{s}|(L_0 - L_i)/L_0|, 1) \in [0, 1]$, where $L_0$ is the loss obtained at $\boldsymbol{x}$, $L_1, ..., L_s$ are the loss values computed on the perturbed samples, and the $\min$ operator is used to upper bound $V(\boldsymbol{x}) \leq 1$. We then set $I_2(\boldsymbol{x}) = 1$ if $V(\boldsymbol{x}) \geq \tau$ (and 0 otherwise). In our experiments, we set the parameters of this indicator as $s = 100$, $r = 10^-3$ and $\tau = 10\%$, as explained in Sect. 3.

*Attack-optimization Failures, Mitigations, and Indicators.* The second category of failures is connected to problems encountered when running a gradient-based attack to solve Problem (1)-(2) with Algorithm 1. This category encompasses failures $F_3$-$F_6$ in Fig. 3.

**$F_3$: Implementation Errors.** We are the first to identify and characterize this failure, which affects many widely-used attack implementations and past evaluations. In particular, we find that many attacks return the *last* sample of the attack path (line 7 of Algorithm 2.1) even if it is not adversarial, discarding valid adversarial examples found earlier in the attack path ($F_3$ in Fig. 3). This issue affects: (i) the PGD attack implementations [25] currently used in the four most-used Python libraries for crafting adversarial examples, i.e., Foolbox, CleverHans, ART, and Torchattacks (for further details, please refer to A.6); and also (ii) AutoAttack [13], as it returns the initial sample if no adversarial example is found, leading to flawed evaluations when used in transfer settings [15].

**$M_3$: Fix Attack Implementation.** To fix the issue, we propose to modify the optimization algorithm to return the best result. This mitigation can also be automated using the same mechanism used to automatically evaluate the IoAF, i.e., by wrapping the attack algorithm with a function that keeps track of the best sample found during the attack optimization and eventually returns it.

**$I_3$: Silent Success.** We design $I_3(\boldsymbol{x})$ as a binary indicator that is set to 1 when the final point in the path is not adversarial, but an adversarial example is found along the attack path (and 0 otherwise).

**$F_4$: Non-converging Attack.** This failure is reported in [41], noting that the flawed evaluations performed by Buckman *et al.* [6] and Pang *et al.* [28], respectively, used only 7 and 10 steps of PGD for testing their defenses. More generally, attacks may not reach convergence ($F_4$ in Fig. 3) due to inappropriate choices of their hyperparameters, including not only an insufficient number of iterations $n$ (line 4 of Algorithm 1), but also an inadequate step size $\alpha$ (line 5 of Algorithm 1).

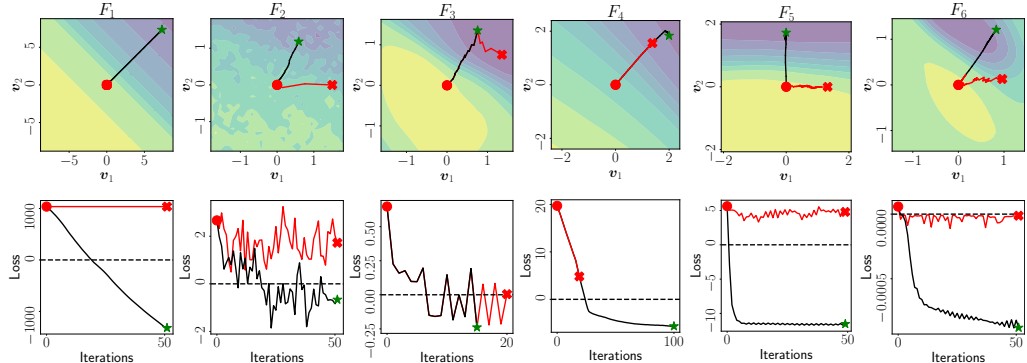

Figure 3: The six attack failures that can be encountered during the optimization of an attack. The failed attack path is shown in *red*, while the successful attack is displayed in *black*. The point $\boldsymbol{x}_0$ is marked with the *red* dot, the returned point of the failed attack with a *red* cross, and the successful adversarial point with the *green* star. The top row shows the loss landscape, as $L(\boldsymbol{x}+a\boldsymbol{v}_1+b\boldsymbol{v}_2, y_i; \boldsymbol{\theta})$. $\boldsymbol{v}_1$ is the normalized direction $(\boldsymbol{x}_n - \boldsymbol{x}_0)$, while $\boldsymbol{v}_2$ is a representative direction for the displayed case. In the second row we show the value of $L(\boldsymbol{x} + \boldsymbol{\delta}_i, y_i; \boldsymbol{\theta})$ for the evaluated model.

**$M_4$: Tune Step Size and Iterations.** The failure can be patched by increasing either the step size $\alpha$ or the number of iterations $n$.

**$I_4$: Incomplete Optimization.** To automatically evaluate if the attack has not converged, we propose a novel indicator that builds a monotonically decreasing curve $\hat{L}$ for the loss, by keeping track of the best minimum found at each attack iteration (a.k.a. *cumulative minimum* curve, shown in black in Fig. 2). Then, we compute the relative loss decrement $D(\boldsymbol{x})$ in the last $k$ iterations (out of $n$) as: $D(\boldsymbol{x}) = |\hat{L}^{(n)} - \hat{L}^{(n-k)}|/(\max_i \hat{L}^{(i)} - \min_i \hat{L}^{(i)}) \in [0, 1]$ . We set $I_4(\boldsymbol{x}) = 1$ if $D(\boldsymbol{x}) \geq \mu$ (and 0 otherwise). In our experiments, we set the parameters $k = 10$ and $\mu = 1\%$, as detailed in Sect. 3.

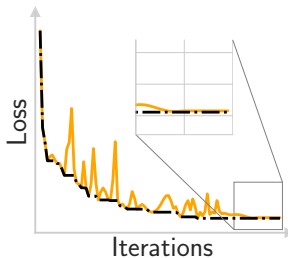

Figure 2: $I_4$ indicator.

**$F_5$: Non-adaptive attack.** This failure is only qualitatively discussed in [3, 11, 41], showing that many previously-published defenses can be broken by developing adaptive attacks that specifically target the given defense mechanism. However, to date, it is still unclear how to evaluate if an attack is *really* adaptive or not (as it is mostly left to a subjective judgment). We try to shed light on this issue below. We first note that this issue arises when existing attacks are executed against defenses that may either have non-differentiable components [16] or use additional detection mechanisms [26]. In both cases, the attack is implicitly executed in a transfer setting against a surrogate model which retains the main structure of the target, but ignores the problematic components/detectors; e.g., Das *et al.* [16] simply removed the JPEG compression component from the model. However, it should be clear that optimizing the attack against such a *bad* surrogate $\hat{\boldsymbol{\theta}}$ (line 2 of Algorithm 1) is not guaranteed to also bypass the target model $\boldsymbol{\theta}$ ($F_5$ in Fig. 3).

**$M_5$: Change Loss (Adaptive).** Previous work [3, 11, 41] applied custom fixes by using better approximations of the target, i.e., loss functions and surrogate models that consider *all* the components of the defense in the attack optimization. This is unfortunately a non-trivial step to automate.

**$I_5$: Transfer Failure.** Despite the failure and some qualitative ideas to implement mitigations being known, it is unclear how to measure and evaluate if an attack is *really* adaptive. To this end, we propose an indicator that detects $F_5$ by evaluating the misalignment between the loss $L$ optimized by the attack on the surrogate $\hat{\boldsymbol{\theta}}$ and the same function evaluated on the target $\boldsymbol{\theta}$. We thus set $I_5(\boldsymbol{x}) = 1$ if the attack sample bypasses the surrogate but not the target model (and 0 otherwise).

**$F_6$: Unreachable Misclassification.** We are the first to identify and characterize this failure. It assumes that gradient-based attacks can get stuck in bad local minima (e.g., characterized by flat regions), where no adversarial example is found ($F_6$ in Fig. 3). When this failure is present, unconstrained attacks (i.e., attacks with $\epsilon \to \infty$ in Eq. 2) are also expected to fail, even if adversarial examples exist for sure in this setting (since the feasible domain includes samples from *all* classes).

**$M_6$: Change Loss (Bad Local Minimum).** We argue that this failure can be fixed by modifying the

loss function optimized by the attack, to avoid attack paths that lead to bad local minima. However, similarly to $M_5$, this is not a trivial issue to fix, and it is definitely not easy to automate.

$I_6$: **Unconstrained Attack Failure.** Despite being difficult to find a suitable mitigation for $F_6$, detecting it is straightforward. To this end, we run an unconstrained attack on the given input $\boldsymbol{x}$, and we set $I_6(\boldsymbol{x}) = 1$ if such an attack fails (and 0 otherwise).

### 2.3  How to Use Our Framework

We summarize here the required steps that developers and practitioners should follow to debug their robustness evaluations with our IoAF.

- *Initialize Evaluation.* The evaluation is initialized with a chosen attack, along with its hyperparameters, number of samples, and the model to evaluate.

- *Mitigate Loss-landscape Failures.* Before computing the attack, the evaluation should get rid of loss-landscape failures, by computing $I_1$ and $I_2$, and applying $M_1$ and $M_2$ accordingly. These indicators are computed on a small random subset of $N$ samples, as the presence of either $F_1$ or $F_2$ would render useless the execution of the attack on all points. Both $F_1$ and $F_2$ are reported if they are detected on at least one sample in the given subset.

- *Run the Attack.* If $F_1$ and $F_2$ are absent or have been fixed, the attack can be run on all samples.

- *Mitigate Attack-specific Failures.* Once the attack completes its execution, indicators $I_3$, $I_4$ and $I_5$ should be checked. Depending on their output, $M_3$, $M_4$, and $M_5$ are applied, and the attack is repeated (on the affected samples). If no failures are found, but the attack is still failing, the value of $I_6$ should be checked, and $M_6$ applied if needed. As $I_1$ and $I_2$, $I_6$ is also computed on a subset of $N$ samples to avoid unnecessary computations. The failure is reported if $I_6 = 1$ for at least one sample.

Let us finally remark that when no indicator triggers, one should not conclude that robustness may not be worsened by a different, more powerful attack. This is indeed a problem of any empirical evaluation, including software testing – which does not ensure that software is bug-free, but only that some known issues get fixed. Our methodology, similarly, highlights the presence of *known* failures in adversarial robustness evaluations and suggests how to mitigate them with a systematic protocol, taking a first concrete step toward making robustness evaluations more systematic and reliable.

## 3  Experiments

In this section, we demonstrate the effectiveness of our approach by re-evaluating the robustness of 7 defenses, and using our IoAF framework to fix them. Additionally, we evaluate 6 more robust image classification models from a widely-used benchmark, discovering that their robustness claims might be unreliable. We also conduct the following additional experiments, reported in the appendix: (i) we describe how we set the thresholds $\tau$ for $I_2$ and $\mu$ for $I_4$ in A.2; and (ii) we offer evidence that IoAF can encompass perturbation models beyond $\ell_p$ norms by analyzing the robustness evaluations of one Windows malware detector in A.3, one Android malware detector in A.4, and one audio keyword-spotting model in A.4.

**Experimental Setup.** We run our attacks on an Intel® Xeon® CPU E5-2670 v3, with 48 cores, 128 GB of RAM, equipped with a Nvidia Quadro M6000 with 24 GB of memory, and we leverage the SecML library [32] to implement our methodology. We show failures of gradient-based attacks on 7 previously published defenses [16, 22, 29, 43, 28, 45, 38, 21, 47], by inspecting them with our IoAF, and most of them [16, 22, 29, 43, 28, 45] are characterized by wrong robustness evaluations. We choose these defenses because it is known that their original evaluations are incorrect [2, 41]; our goal here is to show that IoAF would have identified and mitigated such errors. We include the evaluation of an adversarial detector [38] to highlight that IoAF is able to detect possible failures of this family of classifiers. We include the robustness evaluations of a Wide-ResNet [47] and an adversarially-trained model [21], to highlight that our indicators do not trigger when they inspect sound evaluations. We defer details of the selected original evaluations to A.1. Lastly, to demonstrate that automated attacks do not provide guarantees on performing correct evaluations (without human intervention), we apply the AutoPGD attack [13] with Cross-Entropy loss ($\text{APGD}_{\text{CE}}$) and the Difference of Logits Ratio

Table 1: Indicator values (*cols.*) computed on the selected models (*rows*) using different attacks. The robust accuracy (RA) is reported in the last column (best values in bold). The symbol "Tr" denotes a transfer attack. The ✓ represents the detection of a specific failure. We report in parentheses the fraction of samples for which indicators $I_2$, $I_3$, $I_4$, and $I_6$ are active.

| Model | Attack | $I_1$ | $I_2$ | $I_3$ | $I_4$ | $I_5$ | $I_6$ | RA |
|---|---|---|---|---|---|---|---|---|
| *ST* | **PGD** | | | | | | | **0.00** |
| *ADV-T* | **PGD** | | | | | | | **0.48** |
| *DIST* | **Original** | ✓ | | | | | ✓(10/10) | 0.95 |
| | **APGD**$_{CE}$ | ✓ | | | | | ✓(10/10) | 0.99 |
| | **APGD**$_{DLR}$ | | | | | | | **0.00** |
| | **Patched** | | | | | | | **0.01** |
| *k-WTA* | **Original** | | ✓(10/10) | ✓(23%) | ✓(11%) | | ✓(4/10) | 0.67 |
| | **APGD**$_{CE}$ | | ✓(10/10) | | ✓(21%) | | ✓(2/10) | 0.35 |
| | **APGD**$_{DLR}$ | | ✓(10/10) | | | | ✓(4/10) | 0.28 |
| | **Patched** | | | | ✓(6%) | | ✓(2/10) | **0.09** |
| *IT* | **Original** | | ✓(10/10) | ✓(33%) | ✓(90%) | | | 0.32 |
| | **APGD**$_{CE}$ | | ✓(10/10) | ✓(3% ) | | | ✓(2/10) | 0.12 |
| | **APGD**$_{DLR}$ | | ✓(10/10) | ✓(5%) | | | ✓(1/10) | 0.12 |
| | **Patched** | | | | ✓(3%) | | | **0.00** |
| *EN-DV* | **Original** | | | | ✓(100%) | | ✓(3/10) | 0.48 |
| | **APGD**$_{CE}$ | | | | | | | **0.00** |
| | **APGD**$_{DLR}$ | | | | | | | **0.00** |
| | **Patched** | | | | | | | **0.00** |
| *TWS* | **Original** | | | | ✓(74%) | ✓(6%) | ✓(8/10) | 0.77 |
| | **APGD**$_{CE}$ | | | | | | | **0.03** |
| | **APGD**$_{DLR}$ | | | | | | ✓(7/10) | 0.68 |
| | **Patched** | | | | ✓(1%) | | | **0.01** |
| *JPEG-C* | **Original** | ✓ | | | | | | 0.85 |
| | **Original** (Tr) | | | | | ✓(78%) | | 0.74 |
| | **APGD**$_{CE}$ (Tr) | | | | | ✓(62%) | | 0.51 |
| | **APGD**$_{DLR}$ (Tr) | | | | | ✓(85%) | | 0.70 |
| | **Patched** | | | | | | | **0.01** |
| *DNR* | **Original** | | | | ✓(100%) | | ✓(10/10) | 0.75 |
| | **APGD**$_{CE}$ | | | | ✓(3%) | | | **0.02** |
| | **APGD**$_{DLR}$ | | | | | | ✓(2/10) | 0.20 |
| | **Patched** | | | | ✓(1%) | | | **0.03** |

(APGD$_{DLR}$). Robustness is computed with the *robust accuracy* (RA) metric, quantified as the ratio of samples classified correctly within a given perturbation bound $\epsilon$. For $I_1$, $I_2$, and $I_6$ we set $N = 10$, and for $I_4$ we set $k = 10$. For $I_2$, we set the number of sampled neighbors $s = 100$, and the radius of the $\ell_2$ ball $r = 10^{-3}$, to match the step size $\alpha$ of the evaluations. The thresholds $\tau$ of $I_2$ and $\mu$ of $I_4$ are set to $10\%$ and $1\%$ respectively (details about their calibration in A.2). All the robustness evaluations are performed on 100 samples from the test dataset of the considered model, and for each attack we evaluate the robust accuracy with $\epsilon = 8/255$ for CIFAR models, and $\epsilon = 0.5$ for the MNIST ones. The step size $\alpha$ is set to match the original evaluations (as detailed in A.1).

**Identifying and Fixing Attack Failures.** We now delve into the description of the considered evaluations, which failures we detect and how we mitigate them, reporting the results in Table 1.

*Correct Evaluations (ST, ADV-T).* We first evaluate the robustness analysis of the Wide-ResNet and the adversarially-trained model, by applying PGD with CE loss, with $n = 100$ (number of steps) and $\alpha = 0.03$ (step size). Since no gradient obfuscation techniques have been used, the loss landscape indicators do not trigger. Also, since the attacks smoothly converge, these evaluations do not trigger any attack optimization indicator. Their robust accuracy is, respectively, 0% and 48%.

*Defensive Distillation (DIST)*. Papernot *et al.* [29] use *distillation* to train a classifier to saturate the last layer of the network, making the computations of gradients numerically unstable and impossible to calculate, triggering the $I_1$ indicator, but also $I_6$ indicator as the optimizer can not explore the space. To patch this evaluation, we apply $M_1$ to overcome the numerical instability caused by DIST, forcing the attack to leverage the logits of the model instead of computing the softmax. Such a fix reduces the robust accuracy from 95% to 1%. In contrast, $\text{APGD}_{\text{DLR}}$ manages to decrease the robust accuracy to 0%, as it avoids the saturation issue by considering only the logits of the model, while $\text{APGD}_{\text{CE}}$ triggers the same issues of the original evaluation.

*k-Winners Take All (k-WTA)*. Xiao *et al.* [43] develop a classifier with a very noisy loss landscape, that destroys the meaningfulness of the directions of computed gradients. The original evaluation triggers $I_2$, since the loss landscape is characterized by frequent fluctuations, but also $I_3$, as the applied PGD [25] returns the last point of the attack discarding adversarial examples found within the path. Moreover, due to the noise, the attack is not always reaching convergence, as signaled by $I_4$, also performing little exploration of the space, as flagged by $I_6$. Hence, we apply $M_2$, performing EoT on the PGD loss, sampling 2000 points from a Normal distribution $\mathcal{N}(0, \sigma^2 \mathbf{I})$, with $\sigma = 8/255$. We fix the implementation with $M_3$, and increase the iterations with $M_4$, thus reducing the robust accuracy from 67% to 9%. However, our evaluation still triggers $I_4$ and $I_6$, implying that for some points the result can be improved further by increasing the number of steps or the smoothing parameter $\gamma$. Both $\text{APGD}_{\text{CE}}$ and $\text{APGD}_{\text{DLR}}$ trigger $I_2$, since they are not applying EoT, and they partially activate $I_6$, similarly to the original evaluation. Interestingly, $\text{APGD}_{\text{DLR}}$ converges better than $\text{APGD}_{\text{CE}}$, as the latter triggers $I_4$. Both attacks decrease the robust accuracy to 35% and 28%, a worst estimate than the one computed with the patched attack that directly handles the presence of the gradient obfuscation.

*Input Transformations (IT)*. Guo *et al.* [22] apply random affine *input transformations* to input images, producing a noisy loss landscape that varies at each prediction, thus its original robustness evaluation $I_2$, $I_3$, and $I_4$ indicators, for the same reasons of k-WTA. Again, we apply $M_2$, $M_3$ and $M_4$, using EoT with $\gamma = 200$ and $\sigma = 8/255$, decreasing the robust accuracy from 32% to 0%. Still, for some points (even if adversarial) the objective could still be improved, as $I_4$ is active for them. Both $\text{APGD}_{\text{CE}}$ and $\text{APGD}_{\text{DLR}}$ are able to slightly decrease the robust accuracy to 12%, but they both trigger the $I_2$ and $I_6$ indicator since they are not addressing the high variability of the loss landscape.

*Ensemble Diversity (EN-DV)*. Pang *et al.* [28] propose an ensemble model, evaluated with only 10 steps of PGD, thus triggers the $I_5$ indicator since the attack is not reaching convergence. As a consequence, also the unbounded attack is failing, thus triggering $I_6$. We apply the $M_4$ mitigation and we set $n = 100$, as already done by previous work [41]. This fix changes the robust accuracy from 48% to 0%. Not surprisingly, both $\text{APGD}_{\text{CE}}$ and $\text{APGD}_{\text{DLR}}$ are able to bring the robust accuracy to 0%, since they both automatically modulate the step size.

*Turning a Weakness into a Strength (TWS)*. Yu *et al.* [45] propose an adversarial example defense, composed of different detectors. For simplicity, we only consider one of these, and we repeat their original evaluation with PGD without the normalization operation on the computed gradients, and $n = 50$. This attack does not converge, triggering $I_4$, and if fails to navigate the loss landscape due to the very small gradients, triggering also $I_6$. We apply $M_4$, by setting $n = 100$, and we use the original implementation of PGD (with the normalization step), hence applying also $M_6$, achieving a drop in the robust accuracy from 77% to 1%. For only one point, our evaluation triggers $I_4$, as better convergence could be achieved. Moving forward, $\text{APGD}_{\text{CE}}$ finds a robust accuracy of 3%, as it is modulating the step size along with using the normalization of the gradient. On the other hand, $\text{APGD}_{\text{DLR}}$ triggers the $I_6$ indicator and the robust accuracy remains at 68%, as the attack is not able to avoid the rejection applied by the model.

*JPEG Compression (JPEG-C)*. Das *et al.* [16] pre-process all input samples using *JPEG compression*, before feeding the input to the undefended network. We apply this defense on top of a model with a reverse sigmoid layer [23], as was done in [44]. The authors firstly state that they can not directly attack the defense due to its non-differentiability (thus triggering $I_1$), then they apply the DeepFool [27] attack against the undefended model. This attack has low transferability, as it is not maximizing the misclassification confidence [18]. The robustness evaluation triggers $I_5$ since the attack computed on the undefended model is not successful on the real defense. To patch this evaluation, we apply $M_1$, approximating the back-propagation of the JPEG compression and the reverse sigmoid layers with BPDA [41], and we apply $M_5$ by using PGD to maximize the misclassification confidence of the model. This fix is sufficient to reduce its robust accuracy from

Table 2: Indicator values (*cols.*) computed on the robust models (*rows*), using the $\text{APGD}_{\text{CE}}$ and $\text{APGD}_{\text{DLR}}$ attacks [13]. The robust accuracy (RA) is reported in the last column.

| Model | Attack | $I_1$ | $I_2$ | $I_3$ | $I_4$ | $I_5$ | $I_6$ | RA |
|---|---|---|---|---|---|---|---|---|
| *Stutz* et al. *[39]* | $\text{APGD}_{\text{CE}}$ | | | | | | ✓(10/10) | **0.90** |
| | $\text{APGD}_{\text{DLR}}$ | | | | | | ✓(10/10) | **0.90** |
| *Carmon* et al. *[12]* | $\text{APGD}_{\text{CE}}$ | | | | | | ✓(4/10) | **0.59** |
| | $\text{APGD}_{\text{DLR}}$ | | | | | | ✓(3/10) | **0.55** |
| *Sehwag* et al. *[37]* | $\text{APGD}_{\text{CE}}$ | | | | | | ✓(4/10) | **0.62** |
| | $\text{APGD}_{\text{DLR}}$ | | | | | | ✓(4/10) | **0.57** |
| *Wu* et al. *[42]* | $\text{APGD}_{\text{CE}}$ | | | | | | ✓(4/10) | **0.62** |
| | $\text{APGD}_{\text{DLR}}$ | | | | | | ✓(3/10) | **0.59** |
| *Ding* et al. *[20]* | $\text{APGD}_{\text{CE}}$ | | | | | | ✓(2/10) | **0.47** |
| | $\text{APGD}_{\text{DLR}}$ | | | | | | ✓(2/10) | **0.49** |
| *Rebuffi* et al. *[35]* | $\text{APGD}_{\text{CE}}$ | | | | | | ✓(4/10) | **0.64** |
| | $\text{APGD}_{\text{DLR}}$ | | | | | | ✓(5/10) | **0.65** |

74% to 0.01%. Since APGD can not attack non-differentiable models, we run the attack against the undefended target and we transfer the adversarial examples on the defense. However, both of them trigger $I_5$ as transfer attacks are not effective, keeping the robust accuracy to 51% and 70%.

*Deep Neural Rejection (DNR).* Sotgiu *et al.* [38] propose an adversarial detector encoding it as an additional class that captures the presence of attacks. Similarly to TWS, it was evaluated with PGD with no normalization of the norms of gradients. The attack does not converge, as it triggers $I_4$, but also sometimes gets trapped inside the rejection class, triggering $I_6$. Hence, we apply both $M_4$ and $M_6$, by increasing the number of iterations and considering an attack that avoids minimizing the score of the rejection class. These fixes reduce the robust accuracy from 75% to 3%. Interestingly, only $\text{APGD}_{\text{CE}}$ is able to reduce the robust accuracy to 2%, also noting that it might be still beaten since it is triggering $I_4$. On the contrary, $\text{APGD}_{\text{DLR}}$ is reducing the robust accuracy only to 20%, by also triggering $I_6$. By manually inspecting the outputs of the model, we found that its scores tend to have all the same values, due to the SVM-RBF kernel that assigns low prediction outputs to samples that are outside the distributions of the training samples. This causes the DLR loss to saturate due to the denominator becoming extremely small, making this attack weak against the DNR defense.

**Evaluation of Robust Image Classification Models.** We evaluate 6 defenses recently published on top-tier venues, available through RobustBench [12, 37, 42, 20, 35] or their official repository [39], by applying $\text{APGD}_{\text{CE}}$ and $\text{APGD}_{\text{DLR}}$, using an $\ell_\infty$ perturbation bounded by $\epsilon = 8/255$. We report the results in Table 2. Interestingly, we found that all these evaluations are unreliable, as they trigger the $I_6$ indicator. Hence, even without bounds on the perturbation model, the optimizer is unable to reach regions where adversarial examples are found. This suggests that applying $M_6$, i.e. making the attacks adaptive, can improve the trustworthiness of these robustness evaluations.

## 4  Related Work

**Robustness Evaluations.** Prior work focused on re-evaluating already-published defenses [9, 2, 41, 11], showing that their adversarial robustness was significantly overestimated. The authors then suggested qualitative guidelines and best practices to avoid repeating the same evaluation mistakes in the future. However, the application of such guidelines remained mostly neglected, due to the inherent difficulty of applying them in an automated and systematic manner. To overcome this issue, in this work we have provided a systematic approach to debug adversarial robustness evaluations via the computation of the IoAF, which identify and characterize the main causes of failure of gradient-based attacks known to date. In addition, we have provided a semi-automated protocol that suggests how to apply the corresponding mitigations in the right order, whenever possible.

**Benchmarks.** Instead of re-evaluating robustness of previously-proposed defenses with an ad-hoc approach, as the aforementioned papers do, some more systematic benchmarking approaches have

been proposed. Ling *et al.* [24] have proposed DEEPSEC, a benchmark that tests several attacks against a wide range of defenses. However, this framework was shown to be flawed by several implementation issues and problems in the configuration of the attacks [7]. Croce *et al.* [14] have proposed RobustBench, a benchmark that accepts state-of-the-art models as submissions, and tests them automatically with AutoAttack [13]. The same authors have also recently introduced some textual warnings when executing AutoAttack, which are displayed if the strategy encounters specific issues (e.g., randomness and zero gradients), taking inspiration from what we have done more systematically in this paper. Yao *et al.* [44] have proposed an approach referred to as *Adaptive AutoAttack*, which automatically tests a variety of attacks, by applying different configurations of parameters and objective functions. This approach has been shown to outperform AutoAttack by some margin, but at the expense of being much more computationally demanding.

While all these approaches are able to automatically evaluate adversarial robustness, they mostly blindly apply different attacks to show that the robust accuracy of some models can be decreased by a small fraction. None of the proposed benchmarks has implemented any mechanisms to detect known failures of gradient-based attacks and use them to patch the evaluations, except for the aforementioned warnings present in AutoAttack, and a recent flag added to RobustBench. This flag reports a potentially flawed evaluation when the black-box (square) attack finds lower robust accuracy values than the gradient-based attacks $APGD_{CE}$, $APGD_{DLR}$, and FAB. However, this is not guaranteed to correctly detect all flawed evaluations, as finding adversarial examples with black-box attacks is even more complicated and computationally demanding. Our approach finds the same issues without running any computationally-demanding black-box attack, but just inspecting the failures in the optimization process of gradient-based attacks. For these reasons, we firmly believe that our approach based on the IoAF will be extremely useful not only to debug in a more systematic and automated manner future robustness evaluations, but also to improve the current benchmarks.

## 5    Conclusions and Future Work

In this work, we have proposed the first automated testing approach that helps debug adversarial robustness evaluations, by characterizing and quantifying four known and two novel causes of failures of gradient-based attacks. To this end, we have developed six *Indicators of Attack Failure* (IoAF), and connected them with the corresponding (semi-automated) fixes. We have demonstrated the effectiveness of our methodology by analyzing more than 15 models on three different domains, showing that most of the reported evaluations were flawed, and how to fix them with the systematic protocol we have defined. An interesting direction for future work would be to define mitigations that can be applied in a fully-automated way, including how to make attacks adaptive. Still, our work provides a set of debugging tools and systematic procedures that aim to significantly reduce the number of manual interventions and the required skills to detect and fix flawed robustness evaluations.

To conclude, we firmly believe that integrating our work in current attack libraries and benchmarks will help avoid factual mistakes in adversarial robustness evaluations which have substantially hindered the development of effective defense mechanisms against adversarial examples thus far.

**Acknowledgements**

This work has been partly supported by the PRIN 2017 project RexLearn (grant no. 2017TWNMH2), funded by the Italian Ministry of Education, University and Research; and by BMK, BMDW, and the Province of Upper Austria in the frame of the COMET programme managed by FFG in the COMET module S3AI; by the TESTABLE project, funded by the European Union's Horizon 2020 research and innovation programme (grant no. 101019206); and by the ELSA project, funded by the European Union's Horizon Europe research and innovation programme (grant no. 101070617).

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
