# OpenReview forum: "Indicators of Attack Failure: Debugging and Improving Optimization of Adversarial Examples"
_NeurIPS.cc/2022/Conference — NeurIPS 2022 Accept_

### Official Review · Reviewer_VhaD · 2022-06-27

**Rating:** 6
**Confidence:** 4
**Soundness:** 1 poor
**Presentation:** 3 good
**Contribution:** 1 poor

**Summary:**

The paper provides a set of “quantitative indicators” that can be used to determine if a given adversarial attack has been correctly implemented. By using such indicators, developers of “defenses” can assess whether the attack used to evaluate the defense is “flawed or correct”. The paper also experimentally demonstrates the application of its main proposal, showing that 7 existing defenses are "potentially flawed" due to the incorrect implementation of the corresponding attacks.

**Personal comment**

The paper sets the bar very high by mentioning---many times and starting from the Abstract---the “false sense of security” provided by existing defenses against adversarial examples. Although I liked the initial tone, which intrigued me and induced me to read the rest of the paper with high expectations, I was disappointed by the overall “takeaways” provided by the paper. Indeed, my impression is that the actual contribution of the paper is just a piece of software that can be useful to “debug” evaluations of adversarial attacks---but in the single context of Computer Vision. The paper is full of descriptions and observations, but some of them are either obvious or (apparently) taken by prior work---preventing to determine the “true” contribution to the state of the art provided by the paper. Furthermore, some of the considered defenses have already been found to be "flawed", so I am skeptical about their corresponding "false sense of security" (which is still an overexxageration---discussed below).

Put simply, I believe that the biggest problem of the paper is that it is trying too much. My general recommendation to the authors is to tone down the paper and focus on the specific contribution (which I think is there, but still hidden!) of their work to the state-of-the-art.


**Questions:**

*Incorrect Notation.* The notation of the paper is imprecise. For instance, Equation 1 is ill-defined:

•	“d” is not specified when defining “x”

•	What is “L”? It is only later determined that it is the “loss”, but there is no formal definition.

•	The following statement is vague: “y ∈ {1, . . . , c} is either its label (for untargeted attacks) or the label of the target class (for targeted attacks)”. Actually, it makes no sense: if “y” is “its label” (i.e., the label of “x”) then why would it represent an “untargeted attack”? At most, it should be that “y is any label that is not the correct label of x”.

Reading the rest of Section 2 by assuming the definitions provided by the authors is confusing.
The authors should clearly explain what each term in Equation 1 means. If Eq.1 draws from past work, then the authors should clearly pinpoint which specific work is being used as basis (there are many variations of Eq.1 that have been proposed in the literature, e.g., [A], so it is crucial to pinpoint which work -if any- is being used as basis).


*Only Computer Vision.* Although I understand that NeurIPS is a conference that primarily focuses on Computer Vision (CV) applications of Neural Networks (NN), the fact that the paper only considers attacks (and defenses) on CV is a significant drawback. Indeed, there are much more applications of NN than CV --- all of which have been found to be vulnerable to adversarial attacks (e.g., [B,D]), and all of which having “failures” that do not appear to be considered in this paper (e.g.,[C]); however, this paper only considers CV. What’s worse, is that the main proposal (i.e., the “IoAF”) seems exclusively tailored for the CV domain---which is epitomized by the decision of using SecML, which is a library that also is primarily suited for CV evaluations. In this context, the following statement is misleading: “We also include the evaluation of an adversarial detector [28] to highlight that our framework is able to detect possible failures of this family of classifiers.”: [28] is a “detector” of adversarial examples targeting CV models (indeed, the targeted classifier in [28] are trained on MNIST and CIFAR). To address this weakness, the authors should either (a) consider at least one application of NN which does not pertain to CV, and show that their proposal is effective at identifying failures (if any) also in such domain; or (b) tone down the paper by explicitly mentioning that all the contributions are restricted to CV. Note that the latter will tone down the paper, while the former will increase its impact significantly.


*Exaggerated claims.* The paper states many times that “[the current situation] leads to a ‘false sense of security’”.  From a cybersecurity standpoint, this claim is overexaggerated: such “false sense of security” only exists if either (i) an attack is actually “common” in reality; or (ii) a defense is actually “deployed” in reality. If neither of the two is proven to be true, then the only consequence is that such false sense of security pertains solely to the research domain. I invite the authors to support such a statement with hard facts (justifying either the attack or the defense) or to significantly tone down such a statement.
TO ADD TO THE ABOVE: it is surprising that the paper cites [10], but apparently does not consider a crucial statement of [10], namely: “A large body of work studies a threat model where the adversary is constrained to ℓp-bounded perturbations. This threat model is highly limited and does not perfectly match real-world threat.” Considering that Equation 1 (which is the “backbone” of the paper) clearly refers to lp bounded perturbations, it begs the question of what is the real-world utility of the contribution. Unfortunately, the remainder of Section 2 does not state whether the considered scenario is “restricted” to lp bounded perturbations or can also include completely unbounded perturbations; for instance, the “apply-constraints” function in Algorithm 1 is never defined (I even looked for it in the “supplementary material”). Perhaps the authors can elucidate more on this subject.


*Where is the framework?* Let me quote the first of the claimed contributions reported in the Introduction: “we introduce a unified attack framework that captures the predominant styles of existing gradient-based attack methods, allowing us to identify six main failures that may arise during their optimization (Sect. 2);”. Where, exactly, in Section 2 is the “unified attack framework”? As explained above, the descriptions (algorithm and notation) of Section 2 are vague/unclear, and there are multiple concerns about the generalizability of the main proposal; then, the list of failures in Section 2.1 seems taken from existing literature (most notably [30]). This leads to the next weakness…

*Unclear Contribution w.r.t. SotA?* It is not clear whether some of the “failures/indicators/mitigations” are (i) original content, (ii) reformulations of previous work, or (iii) completely taken from previous work. Sometimes, there are references provided to past work (e.g., F_2 cites [30]) but it is unclear whether such references are used to “support a statement" (e.g., "F_2 is a failure because [reason] which is based on the observations made in [30]”) or to “say that a given 'point' (e.g., a failure) is well-known" (e.g., "F_2 is explicitly stated in [30]"). Note that such a weakness also spans over other parts of the paper. I strongly invite the authors to clearly differentiate original findings/intuitions from well-known “warnings” mentioned by past work. Furthermore, I emphasize that some of such “warnings” are obvious --- especially when considering the mitigations. Consider M3: “if I3 is true, then the implementation is flawed, so the implementation must be fixed accordingly” (but nothing more is said). That’s… common sense? Moreover, such a mitigation is hardly actionable.

*Unsurprising results.* Most of the results are already well known, and hence the experiments, while being useful as a “demonstration”, provide little new insights. For instance, [22] (Distillation) is well known (since 2016) to be inaccurate (it only took a few months after the publication of the paper); it is almost surprising that the paper invalidating [22] is actually referenced in the manuscript (current citation [7]) but used in a completely different context (when explaining F2). Hence (to add to one of my previous weaknesses), what is the “false sense of security” provided by [22]? In this context, a much more "impactful" paper is [30], which thorouhgly elucidates novel issues with 13 defenses (which are much more than the 6 considered in this paper). I understand that the authors have a different objective than simply “invalidating prior defences”, but – as a reviewer and as a reader – the experimental evaluation is detrimental --- in its current form. I invite the authors to either (a) move the evaluation in a dedicated appendix, and then focus on strengthening the true contributions of the paper (which can be exclusively theoretical!); or (b) use the evaluation only to show the most original results (better if they are counterintuitive!).

*Confusing organization.* I strongly believe the paper requires a re-organization that better elucidates the main proposal, due to the fragmentation of the “failures-indicators-mitigations”. Let me explain: the six failures are all listed in Section 2.1. Then, there is Section 3: for each of the six failures, an “indicator” of such failure is proposed; then, in subsection 3.1, there is a mitigation for each failure. Such an organization makes it difficult to “understand” the main proposal: listing all six failures, then all six indicators, and finally all six mitigations, can overwhelm a reader --- especially when it is not needed. Indeed, a quick look at Figure 2 shows that each failure is associated to exactly one indicator, which is then followed by exactly one mitigation. I would have understood the adoption of the current structure if some indicators were shared by more than one failure, or if some mitigations could have been related to more than one indicator/failure. However, considered the complete “1:1:1” mapping, I endorse the authors to better present their contribution: have each failure be followed by its indicator, and then by the corresponding mitigation. Such presentation is much more logical: while reading the paper I had to go back to the previous sections to recall the corresponding failure (or indicator). Potentially, such organization can be further strengthened by integrating a proof-of-concept experiment.

Some lingering issues and suggestions:

•	In Section 5: “our work […] offers a systematic methodology to provide better robustness evaluations”. The “systematic methodology” stands on very brittle foundations (see my main review) and desperately requires an adjustment. My (personal!) impression is that the paper is being oversold: I firmly believe that the contribution is a low-level software toolkit that can be useful in some cases, which is far from being a “systematic methodology”

•	A significant improvement would be listing “all” papers whose evaluations are flawed. Perhaps the search can be restricted only to the papers published in the last few years, and/or published in a few selected venues. Potentially, such analysis can also highlight which papers have been “already” determined to be “flawed” (by other works), and which are found to be “flawed” only by following the proposed set of indicators. Such an analysis would be a far stronger motivation and contribution, and induce almost a straight “accept” from my end.

•	*Where is the “perturbation”?* My intuition is that the adversarial perturbation is always introduced in the “feature space” (according to Equation 1, it is applied to “x” which is the input sample to the ML model). If this is the case, then the approach is not tailored for attacks carried out in the problem space, which are those resembling more realistic scenarios [C]. Of course, a possible response is that "Our focus is on gradient-based attacks, which are typically crafted in the feature space", to which I would reply that "Such attacks have been indicated many times to be difficult to realize in reality". In other words, the restricted focus of the paper is an intrinsic limitation for realisic and "pragmatic" takeaways.

•	*Flawed logic?* In the Introduction, the paper reports that “13 defenses […] were found to be ineffective due to the difficulty of optimizing the attacks correctly [30].” My intuition is as follows: the attacks used to “validate” those 13 defenses were not correctly implemented, hence the “success” (which I assume to be a lower “misclassification rate”) of such defenses were overestimated. If such a reasoning is correct, then it raises the question: the “attack” may not have been correctly implemented, but the misclassifications still appeared so it was (by definition) a “successful” attack---which the defense mitigated. Granted, it was a “different” attack than the one envisioned in the actual paper, but it still induced misclassifications (which were mitigated by the defense). In other words: substantiating the observation that “defenses are ineffective” by stating that “attacks were not correctly optimized” is logically flawed. The only takeaway that can be derived from that logic is that “the defense works against a ‘different’ attack, and the true efficacy of the defense against the ‘expected’ attack is not yet proven”. Indeed, the paper should focus on the implementation of the defense rather than of the attack. I truly invite the authors to clarify this “weakness” and prove me wrong (if I am).


EXTERNAL REFERENCES
[A]: "Adversarial examples for malware detection." European symposium on research in computer security. Springer, 2017.

[B]: "Deep reinforcement adversarial learning against botnet evasion attacks." IEEE Transactions on Network and Service Management 17.4 (2020): 1975-1987.

[C]: "Intriguing properties of adversarial ml attacks in the problem space." 2020 IEEE symposium on security and privacy (SP). IEEE, 2020.

[D]: "Improving robustness of {ML} classifiers against realizable evasion attacks using conserved features." 28th USENIX Security Symposium (USENIX Security 19). 2019

[E]: "Are we learning yet? a meta review of evaluation failures across machine learning." Thirty-fifth Conference on Neural Information Processing Systems Datasets and Benchmarks Track. 2021.

**Limitations:**

An intrinsic (and not reported) limitation is that the work mostly focuses on attacks/defenses in the computer vision domain – neglecting the plethora of other domains in which adversarial examples can be conceived.

Another limitation (which is reported, but still significant and should be a subject of discussion here) is that the proposed set of indicators are only applicable “today”: if new “failures” arise (which are likely, considered the fast advances in this research field) then the current proposal will not be able to detect them --- hence potentially inducing the same sense of “false security” which the authors aim to remove. The issue, here, is that it is a "devil's proof": the contribution shows some indicators---but we cannot now if such indicators can cover all possible causes of "failures" that may affect current adversarial ML evaluations.
Perhaps the authors could attempt to remedy to the abovementioend issue by "promising" to maintain the repository for several years, so that if new failures are found (affecting either current or future researches) then they will be integrated in the toolkit and usable by future researches.
**My concern is this:** the moment a new paper finds a "failure" that is not included in those reported in this paper, then the (currently small) contribution of this paper to the state of the art will be insignificant.

################

UPDATE AFTER INITIAL REBUTTAL: the experiments on malware made me increase my score from 4 to 5, due to "practically proving" that the propose methodology can be adapted to cover also different domains than CV.

UPDATE AFTER AUTHOR'S DISCUSSION: I am increasing the score to a 6 (I would rate it a 6.4)


**Strengths And Weaknesses:**

HIGH LEVEL:

**Originality:** Poor. The only true contribution, in my opinion, are the indicators---which are ~~limited to~~ evaluated on a single application domain (Computer Vision). Investigating "failures" in ML research is not new (e.g., [E]), and many works also do this in the adversarial ML domain (e.g. [30])

**Clarity:** Good. In general, the paper reads well, the motivations are properly explained, and figures/tables are appropriate.

**Quality:** Average. The notation is faulty and the organization can be improved to better reflect the true contribution of the paper.

**Significance:** Poor. Most of the content is well known, and the ~~limited applicability~~ limited evaluation hinders its overall significance to the wide-spectrum of potential ML applications.



LOW-LEVEL

**Strenghts**:
+ Some findings have practical utility for developers
+ The implementation can be useful for future work


**Weaknesses**
- Incorrect notation
- Only Computer Vision
- Exaggerated claims
- “Unsurprising” results
- Where is the framework?
- Unclear contribution w.r.t. SotA
- Confusing organization

REMARK

I thank the authors for their paper: despite my criticism, I endorse their line of research and I would have little objections against this paper if it were submitted to a less prestigious venue.

---

> ### Author Response · Authors · 2022-08-02
> **Answer to VhaD (1/3)**
>
> **Summary.** Before delving into point-to-point responses to the reviewer, we’d like to clarify that a couple of general points which the reviewer reports as criticisms to our paper actually hold more in general for the majority of the papers published in the adversarial machine learning space (approximately more than 5.000 papers to date). Some of them are qualitative arguments that we may also agree with, but that would require an in-depth review of the whole literature of adversarial ML (and we do not believe that this is the right space to address such an important issue - this would require writing a paper on its own supporting some of the reviewer’s statements, and criticizing more than thousands of papers published on top-tier venues), some others are just flawed observations from the technical viewpoint which might reflect a lack of understanding not only of our work but also of the large majority of the literature in adversarial ML.
> In particular, the criticisms about the use of the wording “false sense of security”, the fact that the work focuses on computer-vision models and $\ell_p$-norm attacks, and the confusing remarks about feature-space and problem-space attacks are general criticisms that would invalidate hundreds of attack and defense papers published every year in top-tier venues, including ICML, NeurIPS, ICLR, and computer security conferences such as IEEE SP, USENIX Sec., NDSS, and ACM CCS. Thus, we believe that these observations should neither provide ground for rejecting our work, nor for questioning its suitability for a top-tier venue as those mentioned above.
> Moreover, some of the comments are also wrong from the technical viewpoint. Our work is indeed neither specific to computer-vision models, nor requires using Lp-norm constraints. Whether the attack is implemented in the problem-space or in the feature-space also doesn’t play any role. The only requirement to compute the indicators of failure and use our framework is that the attack is implemented via gradient descent, as also commented more in detail below. We nevertheless thank the reviewer for his/her comments, which we take constructively to clarify some of those unclear aspects in the paper. We have also provided additional experiments to show that our approach is readily applicable outside the computer-vision domain and also when problem-space attacks are considered.
> We do reply in a detailed manner to all criticisms below.
>
> **False sense of security (“exaggerated claims”).** The review starts by criticizing the use of the wording “false sense of security” in our manuscript, which is then also re-discussed later when talking about “exaggerated claims”. We would like to point out that we are not the first to use this notion in our paper. This wording has been introduced by Athalye, Carlini and Wagner in their ICML 2018 paper “Obfuscated gradients give a false sense of security” (which also won the best paper award at the conference), precisely with the same meaning we adopted in our manuscript. This is a well-known way to refer to flawed adversarial robustness evaluations of machine-learning models. So, we respectfully disagree with the reviewer that such wording is exaggerated and inappropriate. We do see the point of the reviewer, on the fact that evidence of gradient-based adversarial attacks and defenses deployed in the wild is debatable and, in this respect, talking about a “false sense of security” might be misleading to a reader who is not quite familiar with the literature of adversarial ML, but this is moving the discussion to a different argument. We would also like to point out that many startups have been created to cope with ML security, providing evidence that the problem is increasingly becoming of practical relevance (see, e.g., Robust Intelligence, BulletProof AI, HiddenLayer, and LatticeFlow). The “false sense of security” as meant in our paper and in other adversarial ML papers refers to the lack of adversarial robustness of ML models against adversarial examples, despite some defenses claiming the opposite. We’ll clarify this definition in the paper, so that the wording does neither sound ambiguous nor exaggerated. Recall also that studying adversarial robustness of ML models has implications which go beyond attacks in the traditional cybersecurity sense. It also serves to study what models learn, debug them, and try to improve the current state of the art in which ML models are just too brittle when compared to how humans perceive and interpret signals. We hope that this clarifies why studying adversarial robustness even against $\ell_p$ norms is of interest to the ML community.

---

> > ### Author Response · Authors · 2022-08-02
> > **Answer to VhaD (2/3)**
> >
> > **Incorrect notation.** We thank the reviewer for this comment, even though notation is not incorrect, but probably just unclear. We will clarify the notation in the updated manuscript (by defining the loss function in formal terms, and clarifying the difference on the role of the label y between untargeted and targeted attacks). We will also explain in the concluding remarks that the indicators of failure can be readily generalized to other domains and perturbation models.
> >
> > **Poor originality/significance.** The reviewer is concerned that investigating failures in ML is not a new problem, citing [E] and that it is widely discussed in the adversarial ML domain. First of all, we do not see any connection with the work [E] mentioned by the reviewer, which points out completely different types of failures occurring when training ML models or implicitly re-using test sets (thereby providing biased performance evaluations). Second, failures of gradient-based attacks have been studied in the adversarial ML literature in [1, 30], but no systematic approach to (1) measure them, and (2) properly tackle them has been ever proposed (besides only guidelines and best practices which have been shown to be largely ignored so far). We invite the reviewer to look, for instance, at the description on how Tramer et al. bypassed kWTA in [30], and evaluate if that approach can be replicated or systematized in any way. All the adaptive attacks that have been demonstrated to show flawed robustness evaluations are neither easily adaptable nor generalizable to other cases, which is also why we keep seeing papers publishing broken defenses. In this respect, our work is the only one which provides a completely novel approach, based on developing 6 indicators of failure (completely original!) that aim to detect wrong evaluations, and suggest how to apply mitigations in the right order to prevent at least the corresponding known failures. We also identified a novel failure and suggested a novel mitigation, useful to patch a flawed PGD implementation in Foolbox. We refer the reviewer to the main response above for further details on the novelty and impact of our work.
> >
> > **Unclear contributions w.r.t. SOTA.** We agree with the reviewer that our contributions have not been clearly stated, especially when disambiguating failures, indicators, and mitigations. We have clarified this point in the main response above.
> >
> > **Only computer vision.** Another main criticism is that our indicators can only be applied to the computer vision domain. This is technically wrong. In fact, our indicators can be applied to any attack which is optimized via gradient descent - an independent requirement from the application domain, and the fact that the attack is crafted in the feature space or in the problem space. Accordingly, our work is applicable to attacks that can be performed via gradient descent also beyond computer vision, including the gradient-based attacks in [C,D]. To demonstrate the generality of our work, we have thus computed our indicators for two distinct attacks computed against Windows and Android malware detectors, showing that no (significant) failure is reported for such evaluations (see the main response above). The reason is that, for the Windows malware detector, the loss landscape is (almost always) smooth and easy to optimize via gradient descent (except on few points for which we observed zero gradients), while for the Android malware detectors (taken from [C]) the gradient is constant and proportional to the feature weights (given that they are linear classifiers! - and, for this reason, it is quite unlikely that the optimization of a gradient-based attack is ever going to fail).
> > We would also like to remark that, whenever feature-space attacks are well modeled (and sometimes a simple combination of $\ell_p$-norms and box constraints can be used for that) they yield a reliable evaluation also when compared to problem-space attacks (cf. the results in [C] and in [a] on the Android malware detectors, which are evaded using the same amount of manipulations) - but again, we point out to the reviewer that this aspect does not influence our work in any way. The indicators of failure are completely agnostic (1) to the choice of the domain, (2) to whether the attack is crafted in the feature or in the problem space, and (3) to the choice of the perturbation model (whether it encompasses $\ell_p$ norms or more general transformation functions). The only requirement to apply them is that gradient-based optimization is used at some point in the process of crafting the attack. Note also that for images (and audio), where additive perturbations are optimized in an end-to-end manner, the feature and problem spaces coincide (as also stated in [C]). This should also clarify the “where-is-the-perturbation” comment of the reviewer.

---

> > > ### Author Response · Authors · 2022-08-02
> > > **Answer to VhaD (3/3)**
> > >
> > > **Unsurprising results.** The reviewer criticizes our work as we have not broken any new defenses, and therefore our results are not surprising. However, again, this is not the goal of our work. The goal of our work is to detect flawed evaluations in an automatic manner, and apply mitigations (following a precise protocol) whenever possible to improve the evaluation. We invite the reviewer to reconsider the paper by evaluating the contribution in this sense. To convince him/her, we have extended our analysis to include more model evaluations, as requested, also including ML-based malware detectors (see the main response above). Interestingly, we found that actually some more recent evaluations are unreliable, and cannot be easily patched. This is captured by our last indicator, which highlights the presence of novel causes of failures, demanding the development of novel adaptive attacks. We do believe that this is a relevant finding, which confirms that our approach is useful and can actually highlight the need of developing novel adaptive attacks or techniques to correctly evaluate defenses (while normally this is only left to the experience of researchers and practitioners, with the catastrophic results that we all know). We will clarify this aspect in the paper, and add the latest experiments presented in this rebuttal.
> > >
> > > **New indicators may be needed.** The reviewer is right when saying that new failures may be discovered in the future, requiring potentially novel indicators and attack mitigations (as we also acknowledge in the previous comment). However, this is not a problem for our approach. As new failures are found, new indicators and mitigations can be added to our approach, which is intrinsically meant to evolve (again, like software testing, when new bugs/vulnerabilities are found, new patches and tests need to be added).
> > > Let us remark again that the indicators work to detect and characterize known failures. The goal of our work is to provide a framework to systematize the detection of known failures and the application of the corresponding mitigations. It is not the scope of this work to find novel causes of failures, novel mitigations, or break new defenses. Conversely, novel indicators should be developed if new failures are found in future work, similar to what happens in software development (e.g., security testing only detects known vulnerabilities - but, even if it is common knowledge, developers still need proper tools to find and patch them). We hope that our response clarifies the scope of this work better and that the reviewer may reconsider its novelty and potential impact under this perspective.
> > >
> > > **Flawed logic.** The reviewer comments that “the observation that defenses are ineffective by stating that attacks were not correctly optimized is logically flawed”, and concludes by saying that defenses were effective against the tested attacks but not against novel adaptive ones. This is again a factual mistake of the reviewer. Many defenses are evaluated against attacks that only run for 10-20 iterations, which do not even converge to a local minimum. These can be shown to be broken by just increasing the number of iterations of the attack, but using the same attack algorithm. The same holds for attacks that return the last point (potentially non-adversarial) but not the best one (which may be adversarial). Some other defenses obfuscate gradients and thus break the attack optimization (but do not improve robustness as instead claimed in the original papers!). Then they’re proven broken by just approximating the model with a smoother function (but again using the same attack). We refer the reviewer to the ICML 2018 paper “Obfuscated gradients give a false sense of security”, as this criticism applies again to all those papers showing that defenses have been wrongly evaluated. Here, wrongly evaluated means that such defenses were tested against attacks that could not find adversarial examples within the given perturbation model and budget, whereas adversarial examples were there. So it is the defense which is wrongly evaluated, many times by misusing attacks. Recall also that testing against weak or misconfigured attacks doesn’t bring any advancement to the state of the art. One should always test a defense against the strongest possible attack within the given perturbation model and budget. This is why we believe that our work is important. We are the first to provide an automated procedure which can spot flawed evaluations, and hopefully this will stop or at least slow down the cat-and-mouse game between most of the attacks and defenses in the adversarial ML space.
> > >
> > >
> > > [a] A. Demontis, M. Melis, B. Biggio, D. Maiorca, D. Arp, K. Rieck, I. Corona, G. Giacinto, and F. Roli. Yes, machine learning can be more secure! a case study on Android malware detection. IEEE Trans. Dependable and Secure Computing, 16(4):711–724, 2019.

---

> > ### Comment · Reviewer_VhaD · 2022-08-03
> > **Thanks and Disclaimer**
> >
> > I thank the authors for their response and for the improvements made to their submission---which prompted me to increase my score.
> > Before I delve into the specifics of their response, however, I would like to clarify some points that have most likely been misinterpreted---at least according to some statements written in the authors' response.
> >
> > **Disclaimer.**
> > The ultimate goal of my review was to improve the paper---and, specifically in this case, to provide suggestions that could be followed to make the paper a significant addition to NeurIPS.
> > It is my belief that the authors interpreted all my remarks as "reasons to reject the paper". This is wrong: the only part of my review that could be considered as such is the "HIGH LEVEL" one, which reports concerns also raised by other reviewers, i.e., the fact that the true contribution does not emerge, and the confusing presentation of the content. These two issues are significant, and are those that made me grade the submission as a 4 (which is a "borderline reject" and not a complete reject). Indeed, these are the reasons that led me to believe that the (initial) submission did not meet the standards of NeurIPS. On this note, let me quote a sentence that was written at the beginning of my review: "*it [the contribution] is there, but still hidden!*".
> >
> > Everything that comes below the "LOW-LEVEL" section of my review are minor issues, remarks and questions whose purpose was to improve the paper in some way. As a matter of fact: I liked the paper (**I even explicitly 'thanked the authors' in my review!**). For example, I would never reject the paper because of the "problem/feature space" matter (which is reported in the "Questions" section and, specifically, below the "some lingering issues and suggestions"). Indeed, let me directly quote from the NeurIPS [reviewer guidelines](https://neurips.cc/Conferences/2022/ReviewerGuidelines): "*Please list up and carefully describe any questions and suggestions for the authors. Think of the things where a response from the author can **change your opinion, clarify a confusion or address a limitation**. This can be very important for a productive rebuttal and discussion phase with the authors.*"
> >
> > I do acknowledge, however, that the format of the review was not easy to interpret (funnily enough, all the 4 reviews of this paper have a significantly different format). Hence, let me clearly state that **my review was NOT meant to "*provide grounds for rejection*" that would "*invalidate the paper*"**.
> >
> > Hopefully, the disclaimer above will serve the twofold purpose of (i) reassuring the authors that my review was not 100% negative, and (ii) setup the stage for a constructive discussion that could lead to a positive outcome.
> > [Note that I increased my score from a 4 to a 5 as a result of the rebuttal, and am willing to increase it even further.]

---

> > > ### Comment · Reviewer_VhaD · 2022-08-03
> > > **The root of the problem**
> > >
> > > Given the limited time available for the discussion, I will prioritize those points that (i) can be addressed in 1 week, and that (ii) if addressed will most likely lead to increasing my score.
> > >
> > >
> > > **THE PROBLEM.** It is a *fact* that all reviewers expressed some concerns w.r.t. (1) the [novelty](https://openreview.net/forum?id=Y1sWzKW0k4L&noteId=rMwZxUz4XP) of the paper's main contribution. Furthermore, the (2) [evaluation](https://openreview.net/forum?id=Y1sWzKW0k4L&noteId=zHba9rb1XqX) was also found not to be satisfying by three reviewers. Finally, all reviewers denoted some confusion w.r.t. (3) the way the content is [presented](https://openreview.net/forum?id=Y1sWzKW0k4L&noteId=9MPbORdC4W) in the paper. Let me point out how all these three parts can be improved---from my perspective, and in a way that does not go against what is stated by other reviewers.
> > >
> > > I will make an effort to provide an exhaustive list of suggestions within the next 12 hours.

---

> > > > ### Author Response · Authors · 2022-08-03
> > > > **Acknowledgement**
> > > >
> > > > First of all, we're glad that the reviewer clarified the standpoint of his/her review, which is now clearer, and we thank him/her for the quick response and the effort put in trying to provide more constructive comments in the next 12 hours. We really appreciate that.
> > > >
> > > > We agree also on the fact that several reviewers commented on the lack of novelty and of clarity in some parts of the paper, which is what we tried to clarify in the rebuttal (and that we will reflect on the revised paper too, if the reviewers agree on that). We have also provided a lot of additional experimental results in the rebuttal that we plan to incorporate in the paper.
> > > >
> > > > But we're looking forward to hearing more from reviewer VhaD, and we will do our best to apply the recommended revisions in due time. Thanks!

---

> > > > > ### Comment · Reviewer_VhaD · 2022-08-03
> > > > > **Ack**
> > > > >
> > > > > I am also glad that the authors acknowledged my response, as well as the underlying purpose of my review. I do sympathyze with them: we're all on the same boat---at different sides, yes, but with a common goal.
> > > > >
> > > > > Nevertheless, I invite the authors to wait before responding to my messages, as I may fix/improve them in the next hours. I will reply to this message once I presented all of my remarks, which will mark the moment after which I will wait for the authors' response.

---

> > > > > > ### Comment · Reviewer_VhaD · 2022-08-03
> > > > > > **Free to go, and some last remarks**
> > > > > >
> > > > > > I have finished summarizing my suggestions. I have made slight edits to all messages, starting from the "[Root of the Problem](https://openreview.net/forum?id=Y1sWzKW0k4L&noteId=31I-ZvEOnC7)" one. I invite the authors to read everything (again, if they already read some messages before I made these edits) and carefully elaborate their response. For completeness, I have also revised my two incorrect statements in my initial review.
> > > > > >
> > > > > > Let me conclude with three remarks:
> > > > > >
> > > > > > * It is up to the authors to decide what to do with my suggestions. Such suggestions are an attempt to improve the paper by (i) enhancing and (ii) elucidating its overall contributions to the SotA. Currently, I am grading the paper as a 5/10, but the paper has high improvements margins and I strongly believe that it can reach an 8---of which I report the corresponding [description](https://neurips.cc/Conferences/2022/ReviewerGuidelines): "*Technically strong paper with, with novel ideas, excellent impact on at least one area of AI or high-to-excellent impact on multiple areas of AI, with excellent evaluation, resources, and reproducibility, and no unaddressed ethical considerations.*"
> > > > > >
> > > > > > * Coincidentally, on line 220 the text says that the experimental platform has 126GB of RAM. That is a rather unusual number: if it is not a typo, care to explain why?
> > > > > >
> > > > > > * Out of curiosity, on lines 346--348 the text says "Here we imagine that our framework could be used to help these tools automatically
> > > > > > detect when their evaluations are incomplete so that they could flag potential errors that should be investigated." How feasible do you think this would be?

---

> > > > > > > ### Author Response · Authors · 2022-08-05
> > > > > > > **Follow-up Answer to VhaD**
> > > > > > >
> > > > > > > We thank the reviewer once again for giving us the possibility to improve our work. We will do our best to upload a revision of the paper which clarifies novelty and improves presentation by Aug. 9. We have also completed the experiments on the audio domain, as reported below. We will add all the completed, additional experiments on RobustBench models, malware detectors, and on the audio domain to the appendix of the paper upon acceptance (before the camera-ready deadline). We hope that this matches the reviewer’s expectations. We reply below to the reviewer’s extensive comments on novelty, presentation, and evaluation.

---

> > > > > > > > ### Author Response · Authors · 2022-08-05
> > > > > > > > **Follow-up Answer to VhaD - Novelty (1/2)**
> > > > > > > >
> > > > > > > > **Novelty.** We report below, for each failure, indicator, and mitigation, whether there is any connection with previous work. Unfortunately, there cannot be a direct experimental comparison with the work in [30], as in that paper each defense was broken with a custom approach which is not possible to systematize/automatize, despite the general failures/mitigations that can be abstracted from that paper. Let us finally recall again that our work has a different goal w.r.t. [30], as it aims to provide (i) novel IoAF to automatically detect known failures, and (ii) an appropriate protocol that suggests which mitigations should be applied, and in which order.
> > > > > > > >
> > > > > > > > $F_1$ (*Non-differentiable/zero gradients*): This failure is only qualitatively described in [1] and [7]. In particular, the evaluation of [1] discusses which qualitative behaviors might imply the presence of this failure; and [7] discusses why the attack used to evaluate the defense is not working.
> > > > > > > >
> > > > > > > > $I_1$: Neither [1] nor [7] proposed a quantifiable indicator to detect the failure, as done in our work. Thus, indicator $I_1$ is original and novel.
> > > > > > > >
> > > > > > > > $M_1$: The BPDA mitigation has been introduced in [1], and also used in [30]. We just suggest applying that when the indicator $I_1$ triggers, as the very first fix to an evaluation.
> > > > > > > >
> > > > > > > > ---
> > > > > > > >
> > > > > > > > $F_2$: (*Unstable loss landscape*): This failure is described in [1], for randomized defenses, and in [30], for the kWTA defense. In particular, the evaluation of [1] only states that randomized defenses can produce meaningless gradients, while [30] considers an ad-hoc analysis of the decision boundary of kWTA (see Fig. 1 in the arXiv version of [30]).
> > > > > > > >
> > > > > > > > $I_2$: None of the two aforementioned works has proposed a systematic, quantitative indicator to measure the presence of noisy/obfuscated gradients, as we did in our work. Thus, indicator $I_2$ is an original contribution of this paper.
> > > > > > > >
> > > > > > > > $M_2$: The application of Expectation over Transformations (EoT) has been introduced in [2] and also used in [1, 30]. We simply recommend using this mitigation if $I_2$ triggers, as the second step of our evaluation protocol.
> > > > > > > >
> > > > > > > > ---
> > > > > > > >
> > > > > > > > $F_3$: (*Implementation error*): This failure has been found in our work, it is an original contribution of this paper. We have found that some implementations of current attacks do not return the best point found during the optimization. These include: Foolbox’s PGD implementation, which returns the last point (potentially non-adversarial), even when adversarial examples are found during the optimization; and AutoAttack, which returns the initial point if no adversarial example is found (causing failures when AutoAttack is used in transfer mode, as discussed in [13]). We will clarify this in the paper.
> > > > > > > >
> > > > > > > > $I_3$: This indicator is a novel contribution of our work.
> > > > > > > >
> > > > > > > > $M_3$: The mitigation corresponds to returning the best (adversarial) point found along the gradient descent path, which again is a novel contribution of this work.
> > > > > > > >
> > > > > > > > ---
> > > > > > > >
> > > > > > > > $F_4$ (*Non-converging attack*): This failure has been originally identified in [30], which qualitatively discusses the presence of non-convergence, pointing out that PGD attacks with 10 steps are likely to fail.
> > > > > > > >
> > > > > > > > $I_4$: This indicator, aimed to detect non-convergence of gradient-based attacks, has been introduced in our work, and it is thus a novel contribution of this paper.
> > > > > > > >
> > > > > > > > $M_4$: This mitigation has already been adopted in [30], by increasing the number of iterations and the step size of PGD. We simply suggest using this mitigation as the fourth step in our evaluation protocol.

---

> > > > > > > > > ### Author Response · Authors · 2022-08-05
> > > > > > > > > **Follow-up Answer to VhaD - Novelty (2/2)**
> > > > > > > > >
> > > > > > > > > $F_5$ (*Non-adaptive attack*): This failure has been discussed in [1, 10, 30], which found that many previously-published defenses can be broken by developing adaptive attacks specifically targeting the defense at hand. However, no guideline/check is suggested to understand whether the attack can be considered really adaptive or not. This is only left to the experience of the researcher/practitioner running the evaluation. And this is also why, again, many papers proposing defenses (and even containing an “adaptive attack evaluation” section) have been shown to be broken afterwards, by a “better-designed” adaptive attack.
> > > > > > > > >
> > > > > > > > > $I_5$: We are the first to propose a simple, quantitative indicator to evaluate if the attack being performed can be considered really adaptive. This can be achieved by simply comparing the behavior of the attack loss with respect to the loss computed on the outputs of the defended model. If the two losses show a different trend (typically, the attack loss decreases while the model loss does not), then it means that the attack is not adaptive, i.e., not appropriately targeting the correct loss function.
> > > > > > > > >
> > > > > > > > > $M_5$: Some custom mitigations based on adjusting the attack loss have been applied in [1,10,30], but they are not generalizable to all cases, and cannot be systematized. Even if we still don’t know how to provide a more systematic approach to implement this mitigation, at least our indicator is able to detect the failure, and flag the evaluation as potentially unreliable.
> > > > > > > > >
> > > > > > > > > ---
> > > > > > > > >
> > > > > > > > >
> > > > > > > > > $F_6$ (*Unreachable misclassification*): This failure has been found in our work, it is an original contribution of this paper. We found that many gradient-based attacks get easily stuck in flat regions or bad local optima that hinder the attack optimization and prevent the attack to find a suitable adversarial example, even when the attack is unconstrained (i.e., when we know for sure that adversarial examples can be found, given that one can ideally replace the input sample with a sample of a different class). The work in [1] has only suggested to run unconstrained and black-box attacks as a sanity check, but nothing more than that.
> > > > > > > > >
> > > > > > > > > $I_6$: We have designed indicator I6 to capture this failure, taking inspiration from one of the sanity checks suggested in [1]. But still, the way $I_6$ is actually computed, is an original contribution of our work.
> > > > > > > > >
> > > > > > > > > $M_6$: Some custom mitigations based on modifying the attack loss function have been shown in [30], but they cannot be generalized to all cases, similarly to $M_5$. In fact, our additional experiments have shown that many models defended with adversarial training trigger this indicator, which potentially highlights that they can be broken by using an attack which explores different paths / optimizes a different loss. However, understanding how to modify the loss in this case is not trivial, and requires experience in designing adaptive attacks (we have some preliminary ideas in this sense which we will continue exploring for future work). Even if we still don’t know how to provide a more systematic approach to implement this mitigation, at least our indicator is able to detect the failure, and flag the evaluation as potentially unreliable.

---

> > > > > > > > > > ### Author Response · Authors · 2022-08-05
> > > > > > > > > > **Follow-up Answer to VhaD - Presentation**
> > > > > > > > > >
> > > > > > > > > >
> > > > > > > > > > **Presentation.** We reply to the points related to the paper presentation below.
> > > > > > > > > >
> > > > > > > > > > Flawed logic. We will add to the paper clear examples of attack optimization failures like those mentioned in the rebuttal (e.g., the attack has been executed for too few iterations to converge). Note that here optimization refers to the execution of the attack algorithm (i.e. the act of maximizing or minimizing a loss function via gradient-based solvers), using a given set of hyperparameters (e.g., step size and number of iterations). We will clarify that in the paper too.
> > > > > > > > > >
> > > > > > > > > > Equation 1. We will clarify both Eq. 1 (defining the loss functions in the targeted and untargeted cases with examples, e.g., using the cross-entropy or the CW loss) and the “apply_constraints” function in Algorithm 1. In particular, the latter is simply meant to represent the projection of the current sample onto the feasible domain. For images, when an $\ell_p$-norm constraint is used (to upper bound the perturbation size by $\epsilon$), along with the box constraint (to ensure that the $d$-dimensional input sample stays, e.g., in $[0,1]^d$), the projection operator takes care of ensuring that both constraints are not violated by the perturbed sample. In the paper, we used $\Delta$ to define the feasible domain (i.e., the intersection of the aforementioned constraints, in the image case).
> > > > > > > > > > In the unbounded case, the $\ell_p$-norm constraint is not used, and thus the projection operator will just enforce the input sample (e.g., the image) to stay within the box constraint (to ensure that pixel values are in the valid range).
> > > > > > > > > > We’d also like to point out that, as shown for the malware case, our approach can work for (i) perturbations that are more generic than just additive ones, and (ii) for perturbation models that are more generic than $\ell_p$-norm constraints, given that both these requirements do not hinder the use of gradient-based approaches. We will clarify this aspect in the paper too.
> > > > > > > > > >
> > > > > > > > > > *False sense of security.* We understand the concern of the reviewer, and to reach a good compromise, we propose to replace “false sense of security” with “false sense of robustness” in the paper. We will also clarify that this refers specifically to (flawed) evaluations which overestimate adversarial robustness.
> > > > > > > > > >
> > > > > > > > > > *Proposal.* If the reviewer agrees, we would like to accept his/her suggestion and retain the term framework (when referring to our approach as a whole) and protocol (when referring to the sequential application of mitigations), while also increasing the use of the IoAF acronym (we will replace most of the occurrences of methodology/protocol/procedure/tool which are unclear from the context).
> > > > > > > > > >
> > > > > > > > > > *Organization.* We will restructure Sections 2.1 and 3 to describe each failure, indicator and the corresponding mitigation within the same subsection. For each of them, we will also clarify what is novel or drawn from previous work, as reported in the novelty comment above.
> > > > > > > > > >
> > > > > > > > > > *Give proper credit to past work.* See our last remark in the “organization” paragraph.

---

> > > > > > > > > > > ### Author Response · Authors · 2022-08-05
> > > > > > > > > > > **Follow-up Answer to VhaD - Evaluation**
> > > > > > > > > > >
> > > > > > > > > > > **Evaluation.** We have run the experiments on the audio domain, considering a well-known keyword-spotting use case, and the results are reported below. Considering additional use cases in which the perturbation model may be different than that used for images (e.g., injection of additional bytes in Windows programs) only requires adapting our $I_2$ indicator. The reason is that, to compute $I_2$, we need to consider sampling some perturbed versions of the input sample, which clearly depends on the perturbation model used. In the malware case, we need to inject random bytes and evaluate the model’s predictions. Once the perturbation function is given/implemented (which is anyway needed to run a meaningful attack, also in the problem space), this only requires a trivial adaptation of the code to compute $I_2$ (implementing the subroutine that perturbs the input samples). The other indicators do not require any modification and can be applied as they are. We hope this clarifies the reviewer’s concerns on the evaluation and on how easy it can be to integrate our approach into other libraries and extend it to other application domains.
> > > > > > > > > > >
> > > > > > > > > > > *Keyword-spotting use case.* We applied the indicators to the audio domain, using a reduced version of the Google Speech Commands Dataset that includes the 4 keywords ‘up’, ‘down’, ‘left’, and ‘right’. We first converted the audio waveforms to spectrograms, then used these spectrograms to train a ConvNet that achieves 99 % accuracy on the test set. The spectrograms were then perturbed in the feature space using the PGD $\ell_2$ attack (in the feature space), and transformed back to the input space using the Griffin-Lim transformation. The samples were then transformed again and passed through the network to ensure the attack still works after the reconstruction of the perturbed waveform. The indicators did not require any change w.r.t. the version used for the image domain.
> > > > > > > > > > > | Table A. IoAF values computed on the Keyword-Spotting use case, using the PGD ($\ell_2$) attack. |  |  |  |  |  |  |  |
> > > > > > > > > > > |---|---|---|---|---|---|---|---|
> > > > > > > > > > > |  | $I_1$ | $I_2$ | $I_3$ | $I_4$ | $I_5$ | $I_6$ | R.A. |
> > > > > > > > > > > | PGD | 0.00 | 0.00 | 0.00 | 0.00 | 0.00 | 0/10 | 0.00 |

---

> > > > > > > > > > > > ### Author Response · Authors · 2022-08-05
> > > > > > > > > > > > **Follow-up Answer to VhaD - Remaining issues**
> > > > > > > > > > > >
> > > > > > > > > > > > **Integration with existing tools/libraries.** We do believe that it is feasible. The IoAF framework is readily applicable to secml, and we plan to integrate it within FoolBox as well. There is also an ongoing integration within IBM ART, and AutoAttack has already implemented some of our indicators.
> > > > > > > > > > > >
> > > > > > > > > > > > **Minor issues.** The workstation has 128GB RAM. We will fix the typo, thanks for spotting it.

---

> > > > > > > > ### Comment · Reviewer_VhaD · 2022-08-05
> > > > > > > > **Looking forward**
> > > > > > > >
> > > > > > > > **Novelty.**
> > > > > > > > Excellent. I hope the authors agree that presenting their IoAF in such a way is much more "convincing" (and fair) than in the original submission.
> > > > > > > >
> > > > > > > > **Presentation.**
> > > > > > > > * *Equation 1.* Perfect.
> > > > > > > > * *False sense of security.* A very elegant way of resolving the issue. Bravo.
> > > > > > > > * *Proposal.* Agreed.
> > > > > > > > * *Organization & Credit.* See above.
> > > > > > > >
> > > > > > > > **Evaluation.**
> > > > > > > > I greatly appreciate this additional experiment.
> > > > > > > >
> > > > > > > >
> > > > > > > > IN SUMMARY: I approve of all these changes, and I look forward to seeing them integrated into an updated version of the paper. My score *will* substantially increase if the authors do a good job.

---

> > > > ### Comment · Reviewer_VhaD · 2022-08-03
> > > > **[1] Novelty**
> > > >
> > > > **NOVELTY**.  First, let me point out that my mentioning of [E] was with respect to the general context of ML, as every year there are increasingly more papers that show "issues" in the evaluation methodologies adopted in research. Indeed, I quote myself "*investigating 'failures' in ML research is not new ([E]) and many works also do this in adversarial ML [30]*". My intent was, again, endorsing the authors in making their contribution better emerge. In our case, this entails having this paper become a valuable contribution to the SotA, which will be actually used and not "forgotten". Such an objective can be reached by **directly comparing with strong-contenders**.
> > > >
> > > > All reviewers unanimously pointed out the many overlaps with this paper and [30]. Hence, I invite the authors to do a 'point-to-point' comparison between their paper and [30]. Indeed, the way the paper currently performs such comparison is with line 325, which I quote: "these [8,1,30,10] can neither specify the reasons why the attacks fail nor be applied automatically on the tested defense.In contrast, our work is inspired by these suggestions, as it offers a systematic methodology to provide better robustness evaluations, also quantifying why and how attack fails". Such a statement was/is (apparently!) insufficient to let the differences/improvements emerge. The key terms, here, are *better* and *systematic*. On this note, a far more convincing argument is what the authors wrote in their response to my review, i.e.: "We invite the reviewer to look, for instance, at the description on how Tramer et al. bypassed kWTA in [30], and evaluate if that approach can be replicated or systematized in any way" I will be frank: if the authors put such a statement in the paper (perhaps with a slightly different wording), it would be far more convincing than the entire text. Consider line 262, which refers to kWTA: "Hence, we apply M2, performing the EoT on the loss of PGD [30] with γ = 2000 and σ = 8/255,": from such a sentence (and the following parts), it was hardly clear what were the limitations of [30] with respect to identifying and fixing the corresponding failure.
> > > > Bottom line: if this paper is *better* than past work (e.g., [30]), then provide irrefutable evidence!
> > > > Such a comparison can be added to the Appendix (referenced in the paper) at no cost.
> > > >
> > > > Moreover, the "systematic" part can also be enhanced by addressing the [Evaluation](https://openreview.net/forum?id=Y1sWzKW0k4L&noteId=zHba9rb1XqX) point...

---

> > > > ### Comment · Reviewer_VhaD · 2022-08-03
> > > > **[2] Evaluation**
> > > >
> > > > **EVALUATION**.  My recommendations on this point are orthogonal to those of the other reviewers. TL;DR: I am satisfied with the current depth of the evaluation. In contrast, my suggestions have the underlying goal of making the proposal *more appealing* to the research domain. Let me elaborate.
> > > >
> > > > The authors were correct in pointing out that my remark about the "suitability to CV" was technically *in*correct. However, it was a fact that the initial submission only performed their experiments on the CV domain. The argument "our proposal can be applied anywhere gradient-based attacks are conceivable" (which, btw, is also something I acknowledged in my review) is clearly correct---*but the paper did not show that*.
> > > >
> > > > Therefore, showing that the proposal (despite being in principle agnostic of the specific domain) has a wide range of applications -- which go beyond the CV setting -- can significantly improve the impact of the paper to the SotA. I already increased my score from 4 to 5 as a result of the added experiments on Malware. I am willing to go even further if the authors consider a yet-another domain (potentially one in which feature/problem space overlap). Such a wide breadth would greatly benefit the statement that the proposal is "systematic". Note that this can be added at no-cost (aside from carrying out the experiments) in the Appendix.
> > > >
> > > > Moreover, I could not overlook the alacrity shown by the authors (which I commend) in performing the new malware experiments, thereby adapting their method also to other domains. I believe that the paper could be even stronger if the authors emphasized its "portability/modularity"---which could be very well supported by explicitly referencing to the fact that, in a short timespan, they adapted their code to gradient-based attacks in a completely different domain (malware) than the one originally considered in the paper (CV). Now, this is a **surprising result** (at least imho), which would emphasize a yet-another value of the proposal, i.e., its *practical utility* (especially w.r.t. existing tools and papers). *Unfortunately, such "utility" is difficult to convey in a paper*. Let's take line 257 as an example (the kWTA): "*Xiao et al. [31] develop a classifier with a very noisy loss landscape, that destroys the meaningfulness of the directions of computed gradients. The original evaluation
> > > > triggers I2*". As a reader, I cannot judge how much of an effort it is required to go from "Xiao et al. [31]" to "triggers I2".
> > > > Perhaps the authors could provide an educational video showing how quick/easy/simple it is to use the proposed IoAF to debug existing works; alternatively, they can describe the amount of effort it took them to realize the malware experiments from the CV ones. Again, this can be added at no-cost in the paper.
> > > >
> > > > Finally, such added value would compensate (in the long term) the intrinsic limitation of the paper, i.e., that the proposed set of IoAF may be incomplete. Let me quote a statement written in response to my review: "*It is not the scope of this work to find novel causes of failures, novel mitigations, or break new defenses. Conversely, novel indicators should be developed if new failures are found in future work, similar to what happens in software development (e.g., security testing only detects known vulnerabilities - but, even if it is common knowledge, developers still need proper tools to find and patch them). We hope that our response clarifies the scope of this work better and that the reviewer may reconsider its novelty and potential impact under this perspective.*"
> > > > --> Such "novel indicators" will only be developed if the proposal (and its practical implementation) is found to be usable by other researchers. However, in the initial submission, there was little that made me believe that this was the case. Indeed, in my review, I stated that "I firmly believe that the contribution is a low-level software toolkit...": it is up to the authors to provide additional evidence that convinces me that (i) it is not *just* a toolkit, and (ii) even if seen in it that way, it is an *amazing* one (worthy of NeurIPS)!
> > > >
> > > > Do note that these suggestions could very well be "added" to the comparison which I requested w.r.t. the "NOVELTY" issue (e.g., AFAIK, the only papers considered in [30] belong to CV).
> > > >
> > > > I believe that fulfilling all of the above would turn the paper into a true research milestone, which would kickstart novel efforts (at both the research and practical level), thereby further magnifying the contribution of this work (which is far greater than the paper itself). It is up to the authors to decide (i) if and (ii) which recommendations to integrate.
> > > >
> > > > This leads me to the final point: the [Presentation](https://openreview.net/forum?id=Y1sWzKW0k4L&noteId=9MPbORdC4W)

---

> > > > ### Comment · Reviewer_VhaD · 2022-08-03
> > > > **[3] Presentation (1/2)**
> > > >
> > > > **PRESENTATION**.
> > > > In simple terms, most of the content of the paper is presented in a "shallow" manner, which generates confusion in the reader. By "shallow" I mean that some statements are made, but for which no explanation is provided (or, if it is, it only comes much later). The ultimate result is that the reader may be induced in faulty thoughts, which are detrimental. Let me elucidate some examples deriving from my review (and consequent response). I anticipate that all such examples do not impair the "quality", but rather the "clarity" of the paper.
> > > >
> > > > * *Flawed Logic*. In my review, I stated the following: "*In the Introduction, the paper reports that “13 defenses […] were found to be ineffective due to the difficulty of optimizing the attacks correctly [30].” My intuition is as follows: the attacks used to “validate” those 13 defenses were not correctly implemented, hence the “success” (which I assume to be a lower “misclassification rate”) of such defenses were overestimated. If such a reasoning is correct...*". The authors replied that "*This is again a factual mistake of the reviewer. Many defenses are evaluated against attacks that only run for 10-20 iterations, which do not even converge to a local minimum. [...] [some] defenses were tested against attacks that could not find adversarial examples within the given perturbation model and budget, whereas adversarial examples were there.*" I thank the authors for their clarification, but my "factual mistake" was due to my (faulty) intuition which was born from line 28 of the paper, which simply states that "*defenses [...] were found to be ineffective due to the difficulty of optimizing the attacks correctly [30]*". The key term here is "optimizing", which can have a lot of meanings and -- unfortunately -- I do not have [30] completely memorized in my head. This issue can be easily rectified by providing a clear example (e.g., the one in the authors' response, i.e., "the defense was found to be effective... against an attack that did nothing.").
> > > >
> > > > * *Equation 1*: in their response, the authors stated that "*Our work does not require using Lp-norm constraints.*". However, in my review I did not state that the work "requires", but rather that it was unclear whether it can be applied also to unbounded ones. Let me quote myself: "*Equation 1 [...] refers to lp bounded perturbations*" (I derived this from line 52) and "*Unfortunately, the remainder of Section 2 does not state whether the considered scenario is 'restricted' to lp bounded perturbations or can also include completely unbounded perturbations; for instance, the 'apply-constraints' function in Algorithm 1 is never defined*". I acknowledge that the paper CAN be applied to unbounded perturbations, but this answer is explicitly found only on line 176 (i.e., in Section 3). Nevertheless, this issue can be rectified easily by (i) properly explaining Equation 1, and (ii) describing what the "apply constraints" comment refers to in Algorithm 1. Note that clarifications on Equation 1 and Algorithm 1 are also requested by [Reviewer tKpS](https://openreview.net/forum?id=Y1sWzKW0k4L&noteId=q6b-v6toYdq).
> > > >
> > > > * *False Sense of Security*. I am glad that the authors understood what I meant with my remark on using such a statement in their paper. However, I DO NOT accept the "appeal to authority" argument: the fact that other papers used such a statement does not make it any more valid (especially because we cannot exclude the risk that a prior work may be "invalidated"...). Furthermore, we are in a different time than the ICML2018 paper referenced by the authors: at that time, it made sense to alert the community about flaws in adversarial ML defenses; nowadays it is less significant---unless there is some evidence showing that the problem (i.e., the "false sense of security") is *real* rather than relegated to the research domain.
> > > > In an attempt to help the authors in substantiating the inclusion of such a statement in their paper, I am pointing them to [MITRE ATLAS](https://atlas.mitre.org/), which reports a collection of case studies of adversarial attacks on production systems. The authors could use such a resource to point out to *proven attacks* (using gradient-based techniques) against *real systems* that could be countered by one of the many defenses considered in the paper. Such (irrefutable!) evidence would provide a much more convincing argument in favor of the "false sense of security". The authors could also use another source, of course. Bottom line: I will approve the inclusion of such a statement if factual evidence is provided (which would strengthen the paper). If no evidence is provided, I recommend either removing or toning down the use of such a statement (the paper would not lose anything in this case).
> > > >
> > > > (continues below)

---

> > > > > ### Comment · Reviewer_VhaD · 2022-08-03
> > > > > **Presentation (2/2)**
> > > > >
> > > > > The latter points are not included in the authors response, but are still relevant.
> > > > >
> > > > >
> > > > > * *Proposal*. What is still unclear to me is the term used to define the actual "proposal". In the paper, a wide array of terms is used, such as: methodology (9 times), protocol (9 times), framework (10 times), as well as procedure and tool. I believe that such diversity contributed in generating the "confusion" among the reviewers, who could not properly identify the improvements over the SotA. I acknowledge that the authors used all such terms as synonyms; yet, this is a case in which clarity should be preferred over style (imho). Bottom line: stick to one term. I would even favor using "IoAF", which -- surprisingly -- is an acronym used only 4 times in the paper (3 of which anticipated by its full definition).
> > > > >
> > > > > * *Organization*. Something I mentioned in my review is the necessity of having to "jump" several times over different sections of the paper. This concern was also raised by [Reviewer xcQc](https://openreview.net/forum?id=Y1sWzKW0k4L&noteId=lnthNxKjHbM), whom I quote: "need to jump forth and back to look for references for six indicators and fixes". In my review I recommended to unify F with I and M, following the structure presented in Figure 2, with the motivation that "there is a 1:1:1 mapping between F:I:M". This would greatly facilitate understanding the main proposal, and would be of great help in following the subsequent evaluation. The authors did not reply to this remark, but I still endorse them to do something about this. Perhaps, instead of restructuring the paper, they could include a Table in the appendix that better explains the "flow" of IoAF.
> > > > >
> > > > > * *Give proper credit to past work*. This has been brought up by most reviewers, but since the authors did not include it in their response I am repeating myself. When presenting the IoAF, the authors should clearly mention the amount of "inspiration" drawn from past work, and which observations are completely original.
> > > > >
> > > > > Do note that the list above does not conflict with any of the remarks made by the other reviewers, and most of such issues can be trivially fixed. The only one requiring more work is the F:I:M table, which is something that I leave to the authors' discretion.

---

### Official Review · Reviewer_tKpS · 2022-07-06

**Rating:** 5
**Confidence:** 3
**Soundness:** 2 fair
**Presentation:** 2 fair
**Contribution:** 1 poor

**Summary:**

This paper proposes a set of indicators of failure for gradient-based adversarial ML attacks, as well as a set of corresponding potential fixes, with the aim of improving current adversarial robustness evaluations through increasing automation and systematization of the process.

The argument is that current proposed checklists can give a false sense of security as they do not ensure a reliable evaluation.


**Questions:**

It seems that the indicators and mitigations in themselves are not novel. Is this the case?

In the results, the original technique is patched with the proposed mitigations. Was some consideration given to applying the same sort of patches to the APGD technique? (if that makes sense to do)

A more general philosophical question is whether the changes to the attacks from the mitigations are better measuring robustness, as is claimed, or are simply different/improved attacks to work around the defences which stop the initial attacks? i.e. is this improving evaluation of robustness against known attacks or simply funding new attack techniques??

Miscellaneous minor points
- Equation (1) is a bit confusing as depending on whether y is the label or target, the approach is different. The footnote (2) on page 2 attempts to clarify this, but it is still a bit of a muddled exposition.
- The discussion just above section 2.1 suggests Alg 1 should return the perturbation for which the objective function is minimised. Surely also some consideration should be given to perturbations which return a different classification than the base label – as this is usually the goal of the attack, and though correlated with the objective function value may be different.
- Presumably the model parameters referred to in Algorithm 1 should include the model architecture as well, i.e. a surrogate model can be a distinct architecture? Or is this implicit already?
- The text in section 2 talks of various options, including an objective being a combination of L and ||\delta|| with some hyperparameters. Yet algorithm 1 (line 5) does not seem to accommodate this.


**Limitations:**

Limitations addressed.

**Strengths And Weaknesses:**

The paper brings together some reasonable indicators of different failure cases and some reasonable mitigations in each of those cases.

My main issues are
-	The list of issues is almost certainly incomplete
-	The indicators for each issue considered are not proven to guarantee to capture the issue (especially in cases where some case by case threshold is required)
-	The mitigations are not guaranteed to be the only reasonable workaround.
-	There are still hyperparameters to be set/fine-tuned empirically/ad hoc.
-
Overall, while bringing together these indicators and mitigations into as more systematic approach is admirable, there are no particular indicators or mitigations that are novel. All seem to have been considered before (in isolation, admittedly) or are just common sense. Thus this paper seems a reasonable guide for practical evaluation and best practise, or even a taxonomy,  but lacks novelty and impact.

---

> ### Author Response · Authors · 2022-08-02
> **Answer to tKpS**
>
> We thank the reviewer for his/her constructive comments, which give us the opportunity to clarify some misunderstandings. We provide point-to-point responses on weaknesses and questions below.
>
> **No particular indicators/mitigations are novel, the paper provides reasonable guidelines and best practices, but it lacks novelty and impact.** We clarified these issues in the comment we provided in the main response posted above, which we refer the reviewer to. In summary, all the indicator metrics are novel. The idea of debugging the optimization process of gradient-based attacks via specific indicators is completely novel and original. Failures and mitigations are all known except the one referred to as “implementation errors” in our paper, which is novel and led us to discover a faulty implementation of PGD in the widely-used FoolBox library.
>
> **Main issues.** The reviewer comments that (i) the list of issues is almost certainly incomplete, (ii) the indicators for each issue considered are not proven to guarantee to capture the issue, and the mitigations are not guaranteed to be the only reasonable workaround, and (iii) there are still hyperparameters to be set/fine-tuned empirically/ad hoc. We agree with the reviewer on these points, but we do not think they are significant issues, as detailed below for each point.
>
> 1. The list of issues is complete as we believe we covered all the known failure cases from the literature (and would be happy to cover more specific ones if the reviewer has any specific suggestions in this respect). However, it is clear that if new cases of failures are discovered in the future, specific indicators should be developed for them. Indeed, this is the whole idea of the framework, which is similar in spirit to the idea of software testing (and, in particular, regression testing): as novel bugs/issues are found, unit tests are developed to avoid that the issue happens again in the future.
>
> 2. The reviewer is right that there may not be any notion of optimality for the indicators themselves, as well as for the corresponding mitigations. We do believe that the problem is indeed similar to unit testing, in which there’s no unique way to implement the tests (and fixes/patches) themselves.
>
> 3. As for the hyperparameter tuning of the attacks, the indicators only provide some guidelines about whether the step size should be changed, or the number of iterations should be increased. We do not believe that it should be complicated to tune these hyperparameters properly. Recall also that we never claimed that the application of the mitigation strategies is completely automated. Our main contribution is to automate the detection of flawed evaluations, via the indicators of failure, and to provide a clear protocol to apply mitigations (which however still require manual intervention).
>
> **Are you developing new attack techniques?** No, we’re not. The reason is that some mitigations are applied to the model itself (e.g., BPDA makes it smoother, bypassing non-differentiable components), and some others just suggest how to better tune the attack hyperparameters. These do not change at all the attack algorithm. The only mitigation that affects the attack algorithm is the EoT, as this requires computing an averaged gradient.
>
> **Application of mitigations to APGD.** In principle, the mitigations can also be applied to other gradient-based attacks, including APGD. Our goal was however even more challenging, as we wanted to show that even a simpler algorithm like PGD could still be used to evaluate defenses in a reliable manner, provided that the causes of failure are ruled out. To facilitate this process, we developed a novel set of indicators that can detect failures automatically, and suggest how to patch the corresponding failures one by one to properly evaluate defenses, even when using simpler attacks. Our work has been the first to envision this process, and we do believe it will help improve the current robustness evaluations.
>
> **Miscellaneous minor points.**  We thank the reviewer for pointing out unclarities in the notation, which we will properly fix. We will also clarify that returning the best point implies that we’re returning the best adversarial (misclassified) point.

---

> > ### Comment · Reviewer_tKpS · 2022-08-04
> > **Response to authors**
> >
> > I thank the authors for their responses to my questions and to their extensive main response to all the reviewers.
> >
> > My main concerns in the review were a lack of significant novelty and impact. Whilst I think my position has not changed re novelty, in that the approach is a framework drawing together mostly known concepts and which are heuristic rather than in any sense rigorously complete, their arguments show some potential impact to practical evaluations in that it can enable a more systematic approach without as much reliance on deep expertise by practitioners. As such, I will adjust my score from a 3 to a 4.

---

> > > ### Author Response · Authors · 2022-08-04
> > > **Acknowledgement**
> > >
> > > We thank the reviewer for acknowledging our rebuttal and increasing the score. We'd like to point out that:
> > >
> > > 1. we will rewrite several parts of the paper in the next days to clarify novelty and impact, as also requested by the other reviewers, and VhaD in particular (we do believe that the idea of providing a set of quantitative indicators of failure to detect flawed evaluations is totally novel and original, as neither such automated "debugging" approach nor any quantitative indicators have been proposed before, even though we necessarily took inspiration from some previously-known failures to develop our approach);
> > >
> > > 2. we are still running additional experiments on the audio domain as requested by VhaD;
> > >
> > > 3. we have provided some details on the quality of the indicators (i.e., how tight the thresholds are) in the response to reviewer xcQc, which we invite the reviewer to check. While this doesn't show any optimality/rigorous property for them, we do believe it is important to show that they are anyway very accurate.
> > >
> > > We'd like also to point out that, if the reviewer has some more suggestions/comments, we would be more than happy to provide additional details or experiments to convince him/her to further raise his/her score.

---

> > > > ### Comment · Reviewer_tKpS · 2022-08-10
> > > > **Response to authors**
> > > >
> > > > Thank you for the updated version of the paper, in particular the clarification of contributions and the highlighting of novel aspects. The additional evaluations are also acknowledged.
> > > >
> > > > I have, as a consequence, incremented my score.

---

### Official Review · Reviewer_xcQc · 2022-07-11

**Rating:** 6
**Confidence:** 4
**Soundness:** 3 good
**Presentation:** 3 good
**Contribution:** 2 fair

**Summary:**

This paper proposes a systematic protocol for improving adversarial example evaluations against common failures. The authors summarize six common failures and propose quantification metrics and meditations for each failure. Experiments on 7 previously broken defenses show that the proposed protocol could have identified and improved their adaptive attacks.

**Questions:**

My current score is based on the following understanding:
* It is hard to prevent inadvertently weak attacks.
* It is good and useful to have a systematic way of identifying weak attacks.
* The proposed indicators provide a better understanding of how and why some attacks are weak.

I am willing to raise my score if the following concerns are adequately addressed:
* [Originality-Weaknesses-1/2, Significance-Weakness-1]
  * Under each indicator, metric, and fix, clarify that they have been previously discussed in the literature.
  * Please discuss the significance and technical contributions beyond previously known techniques.
* Other weaknesses in the Quality section.

**Limitations:**

Limitations have been adequately discussed in the paper.

**Strengths And Weaknesses:**

### Originality

**Strengths**
* The systematic enforcement of failure checking and improving defense evaluation is novel.
* The proposed protocol is a novel extension and identification of well-known pitfalls in evaluating adversarial example defenses.

**Weaknesses**
* **Most techniques are well-known [1, 30].** While I understand (and agree) that connecting well-known failures and fixes is valuable and proposing this systematic protocol is useful, I am concerned that I could not find many new insights from the current discussion. That is, the current discussion is more like a rigorous summary (in terms of a systematic protocol that is indeed novel) of well-known insights discussed in [1, 30]. Part of my concern is that these failures, indicators, and fixes are not clearly cited aside (by explicitly discussing that they have been previously explored).
  * Failures 1-2 and their fixes are already sufficiently discussed by [1, 30]. Their indicators are also discussed by [11], e.g., through black-box attacks and explicitly checking randomness.
  * Failures 3-6 and their fixes are already sufficiently discussed by [30].
* **The evaluation is very similar to [1, 30].** The attacking procedures of many defenses are very similar to those provided by [1, 30], except for the discussion of indicators. For example:
  * k-WTA follows a similar logic as in [30]
  * IT and JPEG-C follow a similar logic as in [1]

### Quality

**Strengths**
* The evaluation is sound and shows that proposed indicators effectively identify weak attacks and that optimizing towards the proposed metrics can improve the robustness evaluation.

**Weaknesses**
* **Unclear tightness of these indicators.** While I understand that passing these indicators does not mean a strong attack, it is suggested to briefly discuss how “tight” these quantifiable indicators are. That is, how likely a weak attacker (potentially outside the discussed defenses) could inadvertently optimize for these indicators yet result in a weak attack not detected by these metrics. I have these concerns because the hyper-parameters and thresholds are only decided by empirical results of a few defenses.
* **Some robustness is still higher than in previous attacks.** I am curious why some evaluations show much higher robustness than previously attacked by [1, 30]. For example, the authors were not able to reduce the robustness of JPEG-C to 0%, as has been done by [1].
* **On the visualization of evaluations.** At L12, the authors mentioned that the proposed indicators could be used to visualize evaluations. Is this claim missing from the paper? I am not sure if it refers to Figure 1. If so, this part is not clearly discussed.

### Clarity

**Strengths**
* The presentation and visualization are generally good.

**Weaknesses**
* The evaluation part is dense, and the readers might need to jump forth and back to look for references for six indicators and fixes.

### Significance

**Strengths**
* The idea of synthesizing known failures as indicators and providing quantifiable metrics can be useful.
* Although most techniques are well-known, synthesizing them in a systematic protocol seems to be a better way of filtering out weak evaluations.

**Weaknesses**
* **Unclear significance in addition to known techniques.** I could not find many new insights from this paper, which is more like an execution manuscript for well-known adaptive attack techniques in a well-known way that has been previously discussed in [1, 30]. Given that most failures and fixes have already been sufficiently discussed in the literature, the only remaining significance seems to be proposing the six metrics. However, I am not sure if these metrics are significant enough, not to mention that some of them have also been discussed before.

---

> ### Author Response · Authors · 2022-08-02
> **Answer to xcQc (1/2)**
>
> We thank the reviewer for appreciating our work and giving us the possibility of clarifying some issues.
>
> **Failures are well-known / unclear significance.** Let us start by highlighting better what we provide as novel in this work. The reviewer reports that Failures 1-2 and their fixes are already sufficiently discussed by [1, 30], which is correct. Then he/she adds that the corresponding indicators are also discussed by [11]. However, the work by Croce et al. spots flawed evaluations in RobustBench by running a black-box (gradient-free) attack after the gradient-based ones (AutoPGD-CE, AutoPGD-DLR, and FAB), and if the black-box attack finds lower robust accuracy values, they flag the evaluation as unreliable. As discussed in the main response, this is not guaranteed to correctly detect all flawed evaluations, as finding adversarial examples with black-box attacks is even more complicated and computationally demanding. We are essentially able to do the same without running any black-box attack or any additional computations (by means of our last indicator $I_6$). To check for randomness, the work in [11] just verifies if the same input produces different outputs, which is not going to work, e.g., on kWTA (obfuscation is different than randomization, even though they can be mitigated similarly via EoT). Indicators $I_1$ and $I_2$ are thus completely novel, as well as indicator $I_6$. Similar comments are valid for Failures 3-6. Except for failure 3 and the corresponding mitigation, which are completely novel and helped us to spot that the PGD implementation in Foolbox is flawed, the other failures and mitigations are not completely novel, even though they have never been categorized in a systematic framework as we have done in our work. Conversely,  all the proposed indicators/metrics are totally new, along with the proposed protocol for patching evaluations. We refer the reviewer also to the main response above for further details on the novelty and impact of our work, as well as to check the additional experimental results provided.
>
> **Some robustness is still higher than in previous attacks.** We thank the reviewer for pointing out this issue. We have actually found that the JPEG-C model used in our paper is not the same used in [1], but it is the model used in [33] that combines JPEG-C from [a] with the reverse-sigmoid defense from [b]. We have fixed our evaluation by applying BPDA also to the additional layer introduced by the reverse-sigmoid defense, and the robust accuracy of the defended model has decreased from 0.46% to 0%. The updated evaluation has been reported in the revised paper uploaded along with this rebuttal. Note however that a 0.46% difference in robust accuracy is quite negligible, the defense can be considered anyway largely broken (given that the original robust accuracy was larger than 70%).
>
> [a] N. Das, M. Shanbhogue, S.-T. Chen, F. Hohman, S. Li, L. Chen, M. E. Kounavis, and D. Chau. Shield: Fast, practical defense and vaccination for deep learning using jpeg compression. In Proceedings of the 24th ACM SIGKDD International Conference on Knowledge Discovery & Data Mining, 2018
>
> [b] T. Lee, B. Edwards, I. Molloy, and D. Su. Defending against neural network model stealing attacks using deceptive perturbations. In 2019 IEEE Security and Privacy Workshops (SPW)

---

> > ### Author Response · Authors · 2022-08-02
> > **Answer to xcQc (2/2)**
> >
> > **On the visualization of evaluations.** The reviewer is right, this point was unclear, and we will remove that claim from the revised paper.
> >
> > **Unclear tightness of the indicators.** We thank the reviewer for this constructive comment, which we try to address below. First of all, given an input sample, the indicators $I_1$, $I_3$, $I_5$, $I_6$ are Boolean (and deterministic) values. The only indicators requiring thresholding (per sample) are $I_2$ and $I_4$, respectively “unstable loss” and “incomplete optimization”.
> >
> > For , we report a table below showing the mean value of the indicator before thresholding (and its standard deviation) over samples, for each model. As one may notice, models with unstable/noisy gradients exhibit values higher than 0.4 for this indicator, whereas non-obfuscated models exhibit values that are very close to zero. We pick 0.1 as the per-sample threshold in this case, but any other value between 0.1 and 0.3 would be actually fine. This indicator is thus not tight, but we preferred a conservative choice of the threshold given that missing a flawed evaluation would be far more problematic than flagging a non-flawed evaluation.
> >
> >
> > |  | $I_2$ | $I_2$ > 0.1 |
> > |---|---|---|
> > | ST | 0.02162+/-0.00122 |  |
> > | ADV-T | 0.00059+/-0.00003 |  |
> > | DIST | 0.00000+/-0.00000 |  |
> > | k-WTA | 0.40732+/-0.03256 | $\checkmark$ |
> > | IT | 1.00000+/-0.00000 | $\checkmark$ |
> > | EN-DV | 0.00014+/-0.00001 |  |
> > | TWS | 0.00862+/-0.00080 |  |
> > | JPEG-C | 0.03129+/-0.00152 |  |
> > | DNR | 0.00000+/-0.00000 |  |
> >
> > Similarly, for I4, we report a table below showing the mean value of the indicator before the thresholding, for each original evaluation. Here, attacks that use few iterations (EN-DV), and attacks that use PGD without the step normalization (TWS, DNR) trigger the indicator. The evaluations that trigger already $I_2$ are not trusted as it is not worth checking convergence for models that present obfuscated gradients.
> > Again, we used for $I_4$ a very conservative threshold to avoid missing the failure.
> >
> > |  | $I_4$ | $I_4$ > 0.01 |
> > |---|---|---|
> > | ST | 0.00279+/-0.00778 |  |
> > | ADV-T | 0.00000+/-0.00000 |  |
> > | DIST | 0.00000+/-0.00000 |  |
> > | k-WTA | 0.04345+/-0.10144 | not trusted ($I_2$) |
> > | IT | 0.00623+/-0.01869 | not trusted ($I_2$) |
> > | EN-DV | 1.00000+/-0.00000 | $\checkmark$ |
> > | TWS | 0.16835+/-0.10307 | $\checkmark$ |
> > | JPEG-C | 0.00002+/-0.00007 |  |
> > | DNR | 0.06857+/-0.01660 | $\checkmark$ |

---

> > > ### Comment · Reviewer_xcQc · 2022-08-07
> > > **Thank you for your detailed response!**
> > >
> > > I really appreciate the detailed response and additional experiments.
> > >
> > > However, the significance of this work is still the main problem. I think I side with other reviewers on this point, as indicated in my original review *"... more like an execution manuscript for well-known adaptive attack techniques in a well-known way that has been previously discussed in [1, 30]."*
> > >
> > > I will try to provide some constructive suggestions from my perspective. I guess what might highlight the significance more is the following flow, which is also what I have tried (but failed) to understand the novelty when writing my original reviews:
> > > 1. **New Failures.** Emphasize brand new failures that are not previously known.
> > > 2. **New Debugging Framework.** Propose the idea of using quantitative metrics to identify these failures.
> > > 3. **Generalize to Existing Failures.** Extend the framework to include existing failures.
> > > 4. **Generalize to Potential Future Failures.** Briefly discuss how this framework extends to failures discovered in the future.
> > >
> > > It is also suggested to highlight the "quantitative metrics" part more, as this part is the main message & contribution if I understand it correctly. Throwing out the entire framework at once (and deferring these metrics to the end) is very likely to blur the real contribution.
> > >
> > > I have not read through other reviewers' discussions and suggestions very carefully, but will generally agree with them (and raise the score) if the old and new knowledge are clearly separated. Note that even from the response, it is very hard to extract a list of new knowledge clearly.

---

> > > > ### Author Response · Authors · 2022-08-08
> > > > **Thanks**
> > > >
> > > > We agree with the reviewer that the novel parts of our work should be better highlighted. We will do our best to reflect the novelty of the approach and the derivation of the indicators in the revised paper.
> > > >
> > > > Meanwhile, regarding what's novel and what's not, all the indicators/metrics are novel (we will emphasize it). In terms of failures and mitigations, we have already replied in detail to VhaD. So perhaps also reviewer xcQc may find these posts clarifying, if he/she hasn't read them yet:
> > > > - https://openreview.net/forum?id=Y1sWzKW0k4L&noteId=1bzgD0Me7Ye
> > > > - https://openreview.net/forum?id=Y1sWzKW0k4L&noteId=M7V74axpjxk
> > > >
> > > > Once again, thank you for your suggestions and effort, appreciated.

---

### Official Review · Reviewer_xRJ5 · 2022-07-11

**Rating:** 5
**Confidence:** 3
**Ethics Flag:** Yes
**Soundness:** 2 fair
**Presentation:** 3 good
**Contribution:** 3 good

**Summary:**

The paper touches on the problem of evaluating the robustness of models to adversarial examples. The proposed systematic evaluation protocol consists of 6 quantitative indicators of failure (IoFA) and corresponding fixes to address the problems. Gradient-based attack failures (e.g., shattered gradients) are described first and then indicators (e.g., unavailable gradients) are introduced accordingly. From this, methods (e.g., use BPDA) to improve the reliability of robustness evaluation are specified to mitigate loss landscape failures and attack optimization failures. Given positive experimental evidence, the proposed pipeline is claimed to be effective by inspecting 7 previously published defenses. Related work, limitations, and potential future work are discussed. Code and data will be public.

**Questions:**

1). Each failure and indicator is coupled with a solution; however, these solutions sound not trivial and automatic. How to decide on the hyper-parameters or settings to apply the suggested skills?

2). Is there an example showing the before-and-after to underline the function of the proposed pipeline?

3). The information in Figure 2 is minimal. It looks like a repetition of the description in the body. Also, Figure 1 is not clear enough for me to distinguish the difference between gray against black and red dot against a cross.

4). The paper ends with the main experiments. Although it already takes huge efforts to involve 7 defenses, the work is not complete to me. Followed studies are necessary.


**Ethics Review Area:**

["Inappropriate Potential Applications & Impact  (e.g., human rights concerns)", "Privacy and Security (e.g., consent)"]

**Limitations:**

Limitations are discussed in Section 6.

**Strengths And Weaknesses:**

Strengths:
1). The paper focuses on the valuable problem of evaluating and improving adversarial defense, which is critical to the robustness of modern deep learning methods in practice. 2). Each indicator is well-motivated by observed failures and is coupled with suggested fixes. 3). The introduced pipeline is successfully examined to improve existing defense baselines by thoughtful experiments.

Weaknesses:
1). Technically speaking, the contribution of this work is incremental. The proposed pipeline is not that impressive or novel; rather, it seems to be a pack of tricks to improve defense evaluation. 2). Although IoFA is well supported by cited works and described failures, its introduction lacks practical cases, where Figures 1 and 2 do not provide example failures and thus do not lead to a better understanding. 3). The reported experimental results appear to evidence the proposed methods, while there is a missing regarding the case analysis and further studies.

---

> ### Author Response · Authors · 2022-08-02
> **Answer to xRJ5**
>
> We thank the reviewer for appreciating our work and highlighting its strengths. We provide additional comments on weaknesses and questions below.
>
> **On novelty and practical impact.** We respectfully disagree with the reviewer that our work is incremental and of little practical value, and refer him/her to the main response provided above.
>
> **Non-trivial mitigations/solutions.** The reviewer is concerned about how difficult it may be to automatically apply the mitigations. We’d like to point out that some of the mitigations are trivial to apply, while some others are not. This is indeed why, while the detection of flawed evaluations can be automated (which is again the main contribution of our work), the same does not hold when it comes to applying mitigations, which still require some manual intervention. We nevertheless believe that already automating the detection of known failures will be very useful and beneficial to the community, at least to make it easier to spot when defenses are not evaluated in a sufficiently reliable manner (i.e., to help rule out known causes of failure).
>
> **Example showing the effectiveness of the proposed pipeline.**  The application of the proposed pipeline/protocol used to apply mitigations is reported in Table 1, along with the indicator values. For each model, we reported the robustness computed with the original evaluation and obtained using AutoPGD, and how the evaluation changes after the application of the suggested mitigations (following the proposed pipeline/protocol).
>
> **Additional experiments.** The reviewer suggested that more models should be evaluated to make our approach more convincing. To this end, we have run additional experiments involving 6 additional robust models for image classification, and 2 more models for malware detection. We refer the reviewer to the main response above for the detailed results, which unveil novel cases of failure. We do believe that the reported findings are relevant, confirming that our approach is useful and can actually highlight the need of developing novel adaptive attacks or techniques to correctly evaluate defenses (while normally this is only left to the experience of researchers and practitioners, with the catastrophic results that we all know). We will clarify this aspect in the paper, and add the latest experiments presented in this rebuttal.
>
> **Minor issues.** We will improve Figure 1 to make differences more visible in the updated paper. We also plan to further improve both Figure 1 and 2 to better clarify our approach.

---

> > ### Comment · Reviewer_xRJ5 · 2022-08-05
> > **Response to authors**
> >
> > Thanks for the huge efforts made to involve additional experiments and the response!
> >
> > One of my major concerns is regarding solutions that are claimed to be the main contribution of this work. However, as the authors stated, parts of the solutions are not that trivial and automatic. From this, the paper, at its core, seems to be more of a tutorial for not fixing but finding a bug, which leads to a lack of novelty and significance. Overall, I will stand in a similar position as the other reviewers and keep my score.
> >
> > Other than the comments of @Reviewer VhaD, I would suggest summarizing many current scattered findings or bugs into a general shared one and proposing a corresponding practical solution so that there is a clear point to highlight and follow.

---

> > > ### Comment · Reviewer_VhaD · 2022-08-05
> > > **A compromise**
> > >
> > > I think Reviewer xRJ5 is correct: the *mitigations* are not (apparently) **all** fully automatic. In fact, this was something I also stated in my original review (specifically referencing to M3). Nevertheless, this pertains only to M3 and M6, whose description is very vague. For completeleness, I report M3 and M6 as written in the original submission:
> > > * *M3: The attack implementation must be fixed accordingly*
> > > * *M6: The evaluation should be repeated with a better attack strategy that converges to adversarial examples when the hard constraints are removed, like applying the normalization of the gradient norm [28, 33].*
> > >
> > > In other words, according to the original submission, failures concerning the "Loss Landscape" (F1 and F2) can be fixed in a fully automatic way, whereas those concerning the "Attack Optimization" are only 50% automatic. Perhaps the wording in the Abstract could be changed to, e.g. "specific fixes to overcome *most of* such failures". Still, from a purely practical perspective, just receiving a warning is still relevant (imho).
> > >
> > > **However**, after reading the authors' response to one of my comments, I am inclined to believe that M3 and M6 (which are original contributions!) are much more "significant" than what was described in the original paper. I invite Reviewer xRJ5 to go over the authors' response (specifically, [this](https://openreview.net/forum?id=Y1sWzKW0k4L&noteId=1bzgD0Me7Ye) and [this](https://openreview.net/forum?id=Y1sWzKW0k4L&noteId=M7V74axpjxk) post).

---

> > > > ### Comment · Reviewer_xRJ5 · 2022-08-07
> > > > **Response to Reviewer VhaD**
> > > >
> > > > After a closer review of the authors' response and your comments, I agree with you that parts of the mitigations are of greater significance. They are even worth a separate study, while my main point stands on the organization of the paper as a whole. There should not be a linear relationship between the stacking of multiple scattered findings and the contribution of the paper. I'm willing to adjust my score if the updated version correctly reflects your comments and the authors' specific responses. However, I still share the point with the other reviewers that there is a lack of clearness in terms of significance. For example, I would prefer a work that specifically focuses on one of the bugs and an accordingly doable solution.

---

> > > > > ### Comment · Reviewer_VhaD · 2022-08-07
> > > > > **Agreed**
> > > > >
> > > > > I wholeheartedly agree with you, Reviewer xRJ5 --- and specifically with this comment: "*I would prefer a work that specifically focuses on one of the bugs and an accordingly doable solution.*"
> > > > > Indeed, this is the reason why, in my review, I stated that "*I believe that the biggest problem of the paper is that it is trying too much.*" Given that NeurIPS only accepts papers up to 9 pages of length, it is intrinsically difficult (if not right-out impossible) to exhaustively solve all the issues that the paper sets out to do.
> > > > >
> > > > > However, the rebuttal (implicitly and explicitly) made me think that the true contribution of this paper lies in its practicality. I may very well be wrong (hopefully not!), but I believe that even if some analyses may require additional depth, such depth would lead to a decreased (potential) contribution to the SotA. In other words: it would be "just another paper" --- technically sound, perhaps, but whose impact would truly be just "incremental". Simply put, I am looking at the "long-term" benefits that this paper can bring to the SotA.
> > > > >
> > > > > Nevertheless, I too am looking forward to seeing how the authors will enhance the paper with all the insights/clarifications presented during this discussion.

---

> > > ### Author Response · Authors · 2022-08-08
> > > **Clarification**
> > >
> > > Thanks to xRJ5 and VhaD for the interesting discussion. We'd like to clarify that our work is not *"a tutorial for not fixing but finding a bug"*, but rather an **automated process** to find such "bugs" (in adversarial robustness evaluations) - which is a big difference in our opinion. This means that, with respect to previous work, we have defined a set of 6 **automated** metrics (all the indicators are novel) which characterize both known and also novel failures (identified in this work, along with the corresponding mitigations - as also pointed out by VhaD - thanks!).
> > > The application of mitigations is however only an additional contribution of the paper, which cannot be clearly fully automated (as the definition of adaptive attacks). We'll clarify these aspects in the paper.
> > >
> > > We'd also like to remark that our work provides an orthogonal contribution w.r.t. finding a new evaluation problem and the corresponding fix. We are not trying to break defenses with (new) adaptive attacks. We are trying to support researchers and practitioners with an automated process that flags wrong evaluations, so that at least they are aware that there may be a problem with the defense under evaluation.
> > >
> > > This approach has never been demonstrated before in the context of adversarial robustness evaluations (whose reliability, as of today, only depends on the experience of the researchers running them), and we firmly believe that it will have significant impact on the state of the art. If we manage to offer a systematic approach to avoid common mistakes, this will possibly help prevent the publication of many papers proposing flawed defenses (which constitutes a big problem today in the adversarial ML field, as even reviewers cannot properly judge the soundness of defenses, and cannot even suggest reliable ways to evaluate them correctly), and finally help us to focus on the right promising directions to improve the status of ML robustness.

---

### Review · Ethics_Reviewer_4rUu · 2022-08-24

**Recommendation:** No ethical concerns.

**Ethics Review:**

An ethics flag was raised based on: Inappropriate Potential Applications & Impact  (e.g., human rights concerns), Privacy and Security (e.g., consent).
While all adversarial attack papers pose some risk of exploitation, the benefit to the community-at-large outweighs the risks of misuse.
This is shown through Shannon's Maxim.
A thorough ethical review of the paper suggests that the proposed tools would serve to promote ethical conduct in the field by promoting positive societal impacts through the provision of automatic testing and debugging tools as well as utilizing the authors' IoAT (six indicators of attack failure).
The checklist was completed and stated that the purpose is to help improve security evaluations.
Notwithstanding the potential of exploitive misuse, it is an acknowledged ethical issue that the authors have considered and addressed.

---

### Author Response · Authors · 2022-08-02
**Main response (1/2)**

We thank the reviewers for the time and effort taken to review our work, and for providing useful and constructive comments. We reply to some general comments below, and also report some additional experiments as requested by the reviewers.

**Novelty and impact.** Many reviewers highlight that our work has limited novelty and potential impact. We respectfully disagree. We believe that the problem we considered is absolutely relevant, as witnessed by the number of published defenses that are constantly broken immediately after publication [17], and even after the publication of clear guidelines and best practices on how to avoid that in 2019 [31,33]. Recall indeed that defenses published in 2020-2022 have been broken again with known tricks [30]. The reason is that such guidelines are difficult to follow and to be applied in a systematic manner, and this is why they have essentially been largely ignored so far.

We do believe that the aforementioned cat-and-mouse game is fueled by the lack of automated debugging tools to help researchers, developers, and practitioners to detect flawed evaluations (which are apparently caught only by very skilled researchers working at the boundaries of machine learning and computer security). The problem is conceptually very similar to finding bugs and vulnerabilities in source code, where developers clearly need automated testing and debugging tools to be able to spot even known issues.

Our work provides a clear and novel contribution in this direction. To our knowledge, nobody has ever attempted to systematically categorize known failures of gradient-based attacks and provide quantitative indicators for them, with the goal of automating debugging and detection of flawed evaluations. In this respect, our work on defining the indicators of failure (all six of them) is completely novel, as well as the failure related to implementation errors and the corresponding mitigation, which allowed us to find that the implementation of the widely-used PGD attack in Foolbox is flawed (via the indicator $I_3$). Let us also remark that Croce et al. implement a similar failure flag to our indicator I6 in the widely-used benchmark RobustBench, to flag evaluations that are potentially unreliable (see the “AA eval. potentially unreliable” column at https://robustbench.github.io), as well as incorporating some of our indicators directly in the AutoAttack framework. This witnesses the soundness and relevance of our work.

It is also worth mentioning that Croce et al. spot flawed evaluations in RobustBench by running a black-box (gradient-free) attack after the gradient-based ones (AutoPGD-CE, AutoPGD-DLR, and FAB), and if the black-box attack finds lower robust accuracy values, they flag the evaluation as unreliable. However, this is not guaranteed to correctly detect all flawed evaluations, as finding adversarial examples with black-box attacks is even more complicated and computationally demanding. We are essentially able to do the same without running any computationally-demanding black-box attack, but just inspecting the failures in the optimization process of gradient-based attacks, as also shown in the additional experiments we have reported below.

We even took one step further in our work, beyond defining the indicators of failure, and showed that there is a clear connection between the characterized failures and some known mitigation strategies, designing a protocol that can be followed to apply them in the right order. This is again a non-trivial contribution of our paper. We will highlight and clarify all these aspects in the paper.

---

> ### Author Response · Authors · 2022-08-02
> **Main response (2/2)**
>
> **Additional experiments.** We conduct eight additional robustness evaluations by considering six computer-vision models, and two malware detectors (Windows and Android), to demonstrate that our approach is generalizable also to other domains and problem-space attacks.
>
> *Image Classification.* The first group of six models comprises recently proposed defenses on top-tier venues, available through RobustBench [b,c,d,e,f] or their official repository [a]. We evaluate them with AutoPGD-CE and AutoPGD-DLR, using an $\ell_\infty$ perturbation bounded by $\epsilon=8/255$. We report the results in Table A.
> Interestingly, we found that all these evaluations are unreliable and trigger the $I_6$ indicator, and that they cannot be easily patched. In fact, the $I_6$ indicator highlights the presence of novel causes of failures, demanding the development of novel adaptive attacks. We do believe that this is a relevant finding, which confirms that our approach is useful and can actually highlight the need of developing novel adaptive attacks or techniques to correctly evaluate defenses.
>
> | Table A. IoAF values computed on six image classifiers using APGD-CE and APGD-DLR. |  |  |  |  |  |  |  |
> |---|---|---|---|---|---|---|---|
> |  | $I_1$ | $I_2$ | $I_3$ | $I_4$ | $I_5$ | $I_6$ | R.A. |
> | (APGD-CE) Stutz et al. [a] | 0.00 | 0.00 | 0.00 | 0.00 | 0.00 | 10/10 | 0.90 |
> | (APGD-DLR) Stutz et al. [a] | 0.00 | 0.00 | 0.00 | 0.00 | 0.00 | 10/10 | 0.90 |
> | (APGD-CE) Carmon et al. [b] | 0.00 | 0.00 | 0.00 | 0.00 | 0.00 | 4/10 | 0.59 |
> | (APGD-DLR) Carmon et al. [b] | 0.00 | 0.00 | 0.00 | 0.00 | 0.00 | 3/10 | 0.55 |
> | (APGD-CE) Sehwag et al. [c] | 0.00 | 0.00 | 0.00 | 0.00 | 0.00 | 4/10 | 0.62 |
> | (APGD-DLR) Sehwag et al. [c] | 0.00 | 0.00 | 0.00 | 0.00 | 0.00 | 4/10 | 0.57 |
> | (APGD-CE) Wu et al. [d] | 0.00 | 0.00 | 0.00 | 0.00 | 0.00 | 4/10 | 0.62 |
> | (APGD-DLR) Wu et al. [d] | 0.00 | 0.00 | 0.00 | 0.00 | 0.00 | 3/10 | 0.59 |
> | (APGD-CE) Ding et al. [e] | 0.00 | 0.00 | 0.00 | 0.00 | 0.00 | 2/10 | 0.47 |
> | (APGD-DLR) Ding et al. [e] | 0.00 | 0.00 | 0.00 | 0.00 | 0.00 | 2/10 | 0.49 |
> | (APGD-CE) Rebuffi et al. [f] | 0.00 | 0.00 | 0.00 | 0.00 | 0.00 | 4/10 | 0.64 |
> | (APGD-DLR) Rebuffi et al. [f] | 0.00 | 0.00 | 0.00 | 0.00 | 0.00 | 5/10 | 0.65|
>
> *Windows malware detection.* We replicate the evaluation of the MalConv neural network for Windows malware detection [g] from [h]. The setting we consider leverages the "Extend" attack, which manipulates the structure of Windows programs in the problem space, while preserving the intended functionality [h].
> We replicate the same setting described in the paper by executing the Extend attack on 100 samples (all initially flagged as malware by MalConv), and we report its performances and the values of our indicators in Table B. As highlighted by the collected results, this evaluation can be improved by applying the BPDA to the final sigmoid applied on the logits of the model, and also by patching the implementation to return the best point in the path.
> We will implement and include the patched evaluations in the paper.
>
> | Table B. IoAF values computed on MalConv, using the Extend attack [h]. |  |  |  |  |  |  |  |
> |---|---|---|---|---|---|---|---|
> |  | $I_1$ | $I_2$ | $I_3$ | $I_4$ | $I_5$ | $I_6$ | R.A. |
> | Extend | 0.36 | 0.00 | 0.12 | 0.00 | 0.00 | 3/10 | 0.26 |
>
> *Android malware detection.* We replicate the evaluation of the Drebin malware detector [i] from [j]. The considered attack only injects new objects into Android applications, to ensure that feature-space samples can be properly reconstructed in the problem space. We limit the budget of the attack to include only 5 and 25 new objects, and we report the collected results in Table C. The attack is implemented using the PGD-LS [j] implementation from SecML.
> Note that, since the evaluated model is a linear SVM, the input gradient is constant, and proportional to the feature weights of the model. Accordingly, the attack is successfully executed without failures.
>
> | Table C. IoAF values computed on Drebin [i], evaluated with PGD-LS [j] with different perturbation budgets. |  |  |  |  |  |  |  |
> |---|---|---|---|---|---|---|---|
> |  | $I_1$ | $I_2$ | $I_3$ | $I_4$ | $I_5$ | $I_6$ | R.A. |
> | PGD-LS (budget 5) | 0.00 | 0.00 | 0.00 | 0.00 | 0.00 | 0/10 | 0.58 |
> | PGD-LS (budget 25) | 0.00 | 0.00 | 0.00 | 0.00 | 0.00 | 0/10 | 0.00 |

---

> > ### Author Response · Authors · 2022-08-02
> > **References**
> >
> > [a] Stutz, D., Hein, M., & Schiele, B. (2020, November). Confidence-calibrated adversarial training: Generalizing to unseen attacks. In International Conference on Machine Learning (pp. 9155-9166). PMLR.
> >
> > [b] Carmon, Y., Raghunathan, A., Schmidt, L., Duchi, J. C., & Liang, P. S. (2019). Unlabeled data improves adversarial robustness. Advances in Neural Information Processing Systems, 32.
> >
> > [c] Sehwag, V., Wang, S., Mittal, P., & Jana, S. (2020). Hydra: Pruning adversarially robust neural networks. Advances in Neural Information Processing Systems, 33, 19655-19666.
> >
> > [d] Wu, D., Xia, S. T., & Wang, Y. (2020). Adversarial weight perturbation helps robust generalization. Advances in Neural Information Processing Systems, 33, 2958-2969.
> >
> > [e] Ding, G. W., Sharma, Y., Lui, K. Y. C., & Huang, R. (2019, September). MMA Training: Direct Input Space Margin Maximization through Adversarial Training. In International Conference on Learning Representations.
> >
> > [f] Rebuffi, S. A., Gowal, S., Calian, D. A., Stimberg, F., Wiles, O., & Mann, T. A. (2021). Data augmentation can improve robustness. Advances in Neural Information Processing Systems, 34, 29935-29948.
> >
> > [g] Raff, E., Barker, J., Sylvester, J., Brandon, R., Catanzaro, B., & Nicholas, C. K. (2018, June). Malware detection by eating a whole exe. In Workshops at the Thirty-Second AAAI Conference on Artificial Intelligence.
> >
> > [h] Demetrio, L., Coull, S. E., Biggio, B., Lagorio, G., Armando, A., & Roli, F. (2021). Adversarial EXEmples: A survey and experimental evaluation of practical attacks on machine learning for windows malware detection. ACM Transactions on Privacy and Security (TOPS), 24(4), 1-31.
> >
> > [i] Arp, D., Spreitzenbarth, M., Hubner, M., Gascon, H., Rieck, K., & Siemens, C. E. R. T. (2014, February). Drebin: Effective and explainable detection of android malware in your pocket. In Ndss (Vol. 14, pp. 23-26).
> >
> > [j] Demontis, A., Melis, M., Biggio, B., Maiorca, D., Arp, D., Rieck, K., ... & Roli, F. (2017). Yes, machine learning can be more secure! a case study on android malware detection. IEEE Transactions on Dependable and Secure Computing, 16(4), 711-724.

---

### Author Response · Authors · 2022-08-09
**Revised Paper: Summary of Major Changes**

We thank all the reviewers for the insightful discussion. We have uploaded a deeply-revised version of our paper (as shown in the diff file, included after the appendix), whose major changes are discussed below.

**Novelty (All reviewers)**
* We have rewritten the abstract and introduction to better clarify the contributions and to mention the new experiments in Sect. 1.
* We have clarified why the proposed checklist and mitigations of [30] are difficult to apply in Sect. 2.2.
* We have included comparisons with previous work and clarified which failures and mitigations are novel and which are not, by also updating Figure 1 to reflect that.
* We have detailed the presence of the implementation failure ($F_3$) inside four adversarial robustness libraries in Sect.2.2 and appendix A.7.

**Evaluation**
* We have discussed how to set the thresholds for $I_2$ and $I_4$ in appendix A.2, showing that they are not tight (Reviewer xcQc).
* We have included all the additional experiments conducted for the rebuttal, covering six robust image classification models, one Windows malware detector, one Android malware detector, and one audio keyword-spotting model in appendix A.3, A.4, A.5, A.6 (Reviewer xRJ5, VhaD).
* We have fixed the evaluation of JPEG-C (its evaluated robust accuracy is now 1%) in Sect. 3 (Reviewer xcQc).

**Presentation**
* We have rephrased “false sense of security” to “false sense of robustness” throughout all the paper (Reviewer VhaD).
* We have simplified the formalization used in Sect. 2.1 for Eq. 1 and Algorithm 1, to improve clarity (Reviewer VhaD, tKpS).
* We have discussed the failures, indicators, and mitigations together in Sect. 2.2 (Reviewer VhaD).
* We have clarified the definition of  transfer attack and surrogate model in Sect. 2.1 (Reviewer tKpS).
* We have included examples of failures of attack optimization in Sect. 2.2 (comment on flawed logic).
* We have used more frequently the IoAF acronym throughout all the paper (Reviewer VhaD).
* We have explained how to generalize to other perturbation models in Sect. 2.2 (Reviewer VhaD).
* We have improved Fig. 3, by enhancing the visibility of lines and markers in Sect. 2.2 (Reviewer xRJ5).
* We have improved the description of the indicators in Sect.2.2.
* We have included the pseudo code for all indicators in appendix A.7.

---

### Meta-Review · Area_Chair_Kc3P · 2022-08-27

**Recommendation:** Accept
**Confidence:** Less certain

**Metareview:**

First, thank you to the reviewers and authors for an in-depth discussion on the contributions and framing of this paper. There’s no doubt this paper was improved over the course of the rebuttal period. And thank you again to the reviewers for having participated in a discussion to clear up the final points of this paper.

This paper essentially presents a checklist of best practices for adversarial evaluation. Such a contribution would hopefully induce more rigorous evaluation standards around adversarial attack research, which is currently often stuck in this feedback loop of empirical attack and defences. New attacks are easily stopped by slight modifications to defences, but they fail under slightly modified attacks.  The authors propose to categorise various attack failures that plague prior work and propose a method to identify these failures ahead of time. They also suggest mitigations to fix these possible sources of failure. In essence, this paper is a combination of survey, reproduction, and opinion paper all in one.

**Strength**: The main strength of this paper is that the framework developed by the authors is practical. Researchers will be able to use this framework to _check_ the rigor of their empirical investigations, hopefully leading to an increase in standards field-wide. However, they also provide strong empirical results across three domains to validate their method of identifying failures.

**Weakness**: There is no “novel” technical contribution (i.e., no new attack & defence). The paper is more a re-hashing of prior approaches. I assume there will be plenty papers of the first variety (with impressive technical chops) at the conference, however, and think this paper stands on its own a different flavour whose content is worth acceptance.

**Award:**

No

---

### Decision · Program_Chairs · 2022-09-14

Accept